# The choice of Madrid as the capital of Spain by Philip II in the light of the knowledge of his time: A transport network perspective

Federico Pablo-Martí[1,2¤]*, Ángel Alañón-Pardo[3,4], Rafael Myro[3]

**1** SCCS Research Group, University of Alcalá, Alcalá de Hernares, Spain, **2** tGIS Research Group, Complutense University of Madrid, Madrid, Spain, **3** Department of Applied and Structural Economics & History, Complutense University of Madrid, Madrid, Spain, **4** Complutense Institute for International Studies, Complutense University of Madrid, Madrid, Spain

¤ Current address: Facultad de Ciencias Económicas, Empresariales y Turismo, Universidad de Alcalá, Alcalá de Henares, Madrid, Spain

* federico.pablo@uah.es

**Data Availability Statement:** All data used are from public sources. In most cases, download links are provided. In addition, the geographic information elaborated by the authors that appears

## Abstract

The suitability of Madrid as the capital of Spain is analyzed from different perspectives, questioning the belief that this choice was eminently personal or political but lacked economic rationality. The paper analyzes Madrid's advantages over other possible capitals from the point of view of both intrinsic characteristics and those that depend on the transport network, such as the problem of supplies or the impact on the development of the surrounding territory. To deal with these questions it is necessary to consider logistical aspects that require an adjusted view of the existing transport network at that time. Using little-known primary sources and a novel methodology based on Delaunay triangulation, the 16th century Spanish transport network is reconstructed with a much higher level of accuracy than ever before. With this information, two maps are prepared that could be used for logistical analysis from a complex network perspective. The first map evaluates the real effects of the choice using an adjusted representation of the territory whilst the second map aims at avoiding the common fallacy of judging decisions made in the past applying current geographical know-how. This map, constructed with the planimetry of the 16th century, indicates how the somewhat deficient knowledge of Philip II with respect to the geographical reality of the day may have favored the choice of Madrid over Toledo, converting some Mediterranean coastal cities into more attractive options. The choice of Madrid as capital appears to be very reasonable in view of the different criteria used. Regarding supply difficulties, our results depart from traditional inclinations by deliberating the fact that the absence of a port in Madrid does not pose an insuperable problem. The latter is the case given that the advantages of maritime transport are far fewer than those usually considered, with Madrid's geographical position offering significant advantages in terms of road transport.

in the figures is included in the Supporting Information files. DATA SOURCES 62. Villuga PJ. El Reportorio de todos los caminos de España: hasta agora nunca visto en el ql allará qlquier viaje q quiera andar muy pvechoso pa todos los caminantes. [Internet]. Medina del Campo: Pedro de Castro a costa de J. de Espinosa; 1546. 256 p. Available from: http://www.traianvs.net/villuga/ 63. Meneses A de. Memorial õ abecedario de los mas principales caminos de España. Ordenado por Alonso de Meneses correo. Va por abecedario: como por la tabla se vera. Con el camino de Madrid a Roma. Va añadido al cabo el Reportorio de las cuetas: reduzidos los escudos a [Internet]. Primera (? Toledo: Juan de Ayala Cano; 1568. Available from: http://diglib.hab.de/drucke/gi-247/start.htm 68. Hibberd R. Mapping sixteenth century Spanish transportation routes: A GIS approach [Internet]. Idaho State University; 2010. Available from: https://www.academia.edu/4016711/Mapping_Sixteenth_Century_Spanish_Transportation_Routes_A_GIS_Approach 106. D'Ocampo F. Los quatro libros primeros de la Cronica general de España [Internet]. Zamora: Juan Picardo; 1544. 470 p. Available from: https://books.google.es/books?id=1yRhAAAAcAAJ&dq=caminos&hl=es&source=gbs_similarbooks 107. Estienne C. Les voyages de plusieurs endroits de France & encores de la Terre Saincte, d'Espaigne, d'Italie & autres pays: Les fleuves du royaume de France [Internet]. Paris: Chez Charles Etienne; 1552. 113 p. Available from: https://gallica.bnf.fr/ark:/12148/bpt6k327907f.image 109. Stella C Della, L'Herba G da. Poste per diverse parti del mondo. Et il viaggio di S. Giacomo di Galitia, tutte le fiere principali del mondo. La narratione di Roma, e delle sette chiese brevemente ridotta [Internet]. Venice: Domenico Farri; 1564. 200 p. Available from: https://books.google.es/books?id=NDISAAAAcAAJ&printsec=frontcover&hl=es&source=gbs_ge_summary_r&cad=0#v=onepage&q&f=false 110. Rowlands R. The post for diuers partes of the world to trauaile from one notable Citie unto an other, with a description of the antiquitie of diuers famous Cities in Europe [Internet]. London: Thomas East; 1576. 112 p. Available from: https://en.wikisource.org/wiki/The_Post_of_the_World 111. Hogenberg F. Hispania [Internet]. 1579. Available from: https://gallica.bnf.fr/ark:/12148/btv1b55010185x/f17.item 148. Andree L, [Loew (alias M. Quad) C. Kronn und Auszbunde aller Wegweiser. Darinne verzeichnet seindt alle die Wege, so gehen ausz 71. den vornembsten Städten von Teutschland, 17t. von Niderlandt, 39. von Frankreich, 29 von Italia, und 31. von Hispania [Internet]. Colonia: Andree, Lambereum; 1597. 322 p. Available from: https://play.google.com/store/books/details?id=

## Introduction. The unfinished debate concerning the choice of Madrid as capital

During the Middle Ages and until the 16th century, the itinerant court was the usual way in which to govern the kingdoms of Europe. The king strove to be present in all his territories in order to maintain some measure of control. To do this, the king moved not only himself and his family but also the entire court formed by the nobles, soldiers, servants, officials, and a growing archive of different documents. The costs of any relocation were usually borne by the cities in which the court established itself with a longer stay signifying a substantial financial burden. This system became more inefficient and costly as the bureaucratic apparatus grew with the modern states [1–3].

Few historical facts are still the subject of such passionate debate, both academically, politically, and socially, as Philip II's choice of Madrid as his permanent residence and the imperial court in 1561. Whilst the debate continues today in many aspects, it is acknowledged that the king meticulously planned the election of Madrid as the court's permanent residence. Despite other more relevant Castilian cities, he chose Madrid for the purpose of practicing a far-reaching political and cultural program [4]. This decision has had huge repercussions in terms of the subsequent development of the Spanish road network and infrastructure and, consequently, on the spatial configuration of economic activity. The latter is the main reason for the endless debates about Madrid's suitability as the capital [5, 6], as well as its effects on the Castilian hinterland [7–9].

Although there seems to be some consensus regarding the need to establish a capital to adequately rule the immense domains of the Spanish branch of the Habsburg dynasty, the suitability of the choice of the city is more controversial.

Besides Madrid, other cities have been pointed out as possible locations, such as Valladolid, Toledo, Seville, Barcelona, or even Lisbon, following Cardinal Granvelle's advice to better control the Atlantic Ocean [10]. There are multiple and diverse determining factors for the choice of capital. The literature on the subject is consequently very varied and challenging to rank in terms of any common thread.

One approach is to analyze all the relevant factors sequentially; we would expect a decision of this kind, under the assumption that Philip II did not choose the capital by evaluating all possible sites simultaneously but in various stages [11]. At the first stage, the territorial scope would be established: the set of kingdoms that would have the city as their capital. At the second, the advantages and the disadvantages offered by the different areas of the chosen territory would be evaluated. Finally, at a third stage, the capital would be selected, considering the specific characteristics of each city. This selection process by stages would seem to fit Philip II's personality well besides being useful from a practical point of view. According to Zarco Cuevas [12], included in the advice given to his successor on his deathbed, he recommended that the court remain in Spain, but without indicating any specific city, which gives us reason to believe that he considered this issue as secondary:

*"You, King of Spain, must reside in Spain, because, although Italy (former seat and mother of the Empire, located between two seas, neighbor of Africa, and not far from Greece and other countries of the Turk, and almost halfway between Spain and Flanders) borders France and Germany, it will be to your greatest convenience to reside in Spain, where, with you will preside overall, because it will serve you, as a bridge and step for the whole Monarchy. No other country is more used to seeing and having its King present, without whom it is lost, and for the navigation of the Indies and to restrain England, there is no better place than this."* (Philip II, the quotation is taken from Zarco Cuevas [12].

P3RYAAAAcAAJ&rdid=book-P3RYAAAAcAAJ &rdot=1 149. Eichoviothias (alias M. Quad) C. Deliciae hispaniae et index viatorius, indicans itinera, ab vrbe Toleto, ad omnes in Hispani civitates et oppida [Internet]. Ursel: Cornelij Sutorij; 1604. 71 p. Available from: https://books.google. es/books?id=YB5YAAAAcAAJ&hl=es&source= gbs_similarbooks 150. de Mayenr TT. Sommaire description de la France, Allemagne, Italie, Espagne: avec la guide des chemins pour aller et venir par les provinces, et aux villes plus renommés de ces quatre regions [Internet]. J. Stoer; 1605. 351 p. Available from: https://www.e-rara.ch/gep_r/doi/10.3931/e-rara-51022 161. Correas P. Poblaciones españolas de más de 5000 habitantes entre los siglos XVII y XIX. Bol Asoc Demogr Hist [Internet]. 1988;6(1):5–23. Available from: https://dialnet.unirioja.es/descarga/articulo/ 103939.pdf"

**Funding:** This work has been supported in part by Comunidad de Madrid through Grant H2019/HUM-5761 (INNJOBMAD-CM) and Universidad de Alcalá COVID-19 UAH 2021 2020/00003/016/001/003. The funders had no role in study design, data collection and analysis, decision to publish, or preparation of the manuscript.

**Competing interests:** The authors have declared that no competing interests exist.

Concerning the first stage of the decision-making process, it is necessary to determine which territories were likely contenders for the location of the capital in the run-up to 1560, when the decision was taken. This issue is essential yet not easy to solve [13] since there no primary sources exist indicating the list of cities that Philip II considered. Whether only territories within the Iberian Peninsula, or if all his European domains were entertained as possibilities is unknown. As aforementioned, at the end of his reign, Philip's position on the matter seems clear, although there are no documents that allow us to determine with real certainty whether this was the case when the choice of the capital was made. However, it seems most plausible that only the peninsula was considered. When Philip II decided to establish a capital for his empire, likely his attention was more focused on the Iberian territories because his father has previously removed him from the line of succession to the imperial crown in 1531.

Regarding these territories, it seems clear that there were only two options, the kingdoms of Castile and Aragon, since Portugal was not integrated into Philip II's domains until 1580, almost two decades after the decision regarding the capital was taken.

However, there is no consensus on either of the two issues mentioned above, the need to establish a capital for the Hispanic Monarchy and the suitability of Madrid for it. Ringrose [14] considered that the very existence of Madrid as a capital city was the consequence of a political decision. Perhaps one of the authors who has disagreed most radically is Elliot [10], for whom Madrid's only merit was its geographical location. He argues that the choice of capital, central and remote at the same time, contradicted one of the fundamental bases of the Spanish monarchy. If the various territories that formed the monarchy were considered independent units and placed on the same level, they all deserved an identical degree of consideration. Therefore, the establishment of a permanent capital meant the renouncement of the Carolingian practice of the itinerant monarchy, a course which, despite its drawbacks, had the great advantage of occasionally giving its people visible proof that their king had not forgotten them. Believing that he could be in close contact with the needs and problems of his territories from his observation post in the geographical center of Spain, Philip II did not attach importance to the fact that this solution to the problem would prevent his territories from remaining in contact with him. Elliot [10] also pointed out that Philip II was mistaken in believing that the decision to reside in the geometrical center of the peninsula would produce the impression of absolute impartiality in the treatment accorded to his subjects, since, although this was not the intention of Philip II, the very choice of the capital in the heart of Castile granted his government a Castilian color.

Despite the great importance of this historical choice, historians have not studied it in depth from an economic perspective. However, this does not mean that it has not been commented on and discussed a great deal, with religious and political reasons often cited as to why Madrid was more suitable than, say, Valladolid, Toledo, Seville, or even Barcelona or Lisbon.

According to such suggestions, Madrid's main advantage, from Philip II's point of view, was probably that neither the Church nor the nobility enjoyed an important presence there, which would facilitate the development of a new administration, under solely royal supervision. Also, following the Comuneros' Revolt of Madrid, the expropriation of lands meant that the King had a great deal of land in Madrid at his disposal. Moreover, it appears that the city was blessed with a considerable body of civil servants [6].

Madrid's choice would not have been excessively provocative to the two main Castilian cities, Valladolid and Toledo, regarding the balance of power. With the Spanish primate residing in Toledo and the Chancery located in Valladolid, Philip II could take his court to Madrid, a location which he could model and urbanize to his liking. He had undertaken a three-year trip in 1548 to get to know his future European territories, during which he had acquired many architectural and urbanistic ideas, particularly in the Netherlands. Santos Vaquero [15] notes

that it is reasonable to think that he also saw the location of the capital in Toledo as an obstacle to realizing the plans these ideas had inspired.

However, it would appear strange that, Madrid's suitability as capital should not have been considered by Philip II and his advisers from an economic perspective. Nevertheless, despite the issue's importance, it has attracted little direct analysis. One possible reason could be that most historians have taken it for granted [16, 17], not only because of the apparent geographic centrality of Madrid but also because of its economic centrality. The latter is of course what matters, particularly as some influential authors regard Madrid as the communications link not only between the two plateaus but also between the North and South of Spain [18–20]. Besides, Madrid's population doubled from 1500 to 1560 due to the prosperity of agriculture and animal breeding activities, which both benefited from the diversity of rivers and streams. These developments created a relevant market in the city. However, due to the rapid increase in food demand, supply crises were unavoidable in the second half of the 16th century [21].

Nevertheless, from the middle of the 1970s, Madrid's economic centrality started to be strongly questioned by studies not directly focused on analyzing Madrid's choice as a capital. Ringrose, in his influential works of 1973 and 1983, wrote:

*"The very existence of Madrid as a capital city was the consequence of a political decision. No other city in early modern Europe was as dependent upon administered economic life, and no major city was so poorly located to stimulate market-oriented exchanges"* [14].

González Enciso [8] concurred with this, as did Kindleberger, who, quoting Ringrose, compared Madrid with Rome at the time of the Roman Empire:

*"Like Rome in Italy, it was parasitical, housing the court, grandees, hidalgos, and bureaucrats but consuming, in addition to grain for its poor, luxury goods from abroad and semi luxuries from local crafts. It failed to galvanize Castile. The various coasts were more closely tied to one another and foreign countries—Barcelona to the east, Seville to the colonies, Bilbao to France and northern Europe—than to the center of the country."* [22]).

Arguing along the same lines, Gómez Mendoza [23] points out that, "Except Madrid, all the capitals of the European Union are located a short distance from the sea or on the banks of large navigable rivers. The geographical isolation of Madrid marked its historical evolution and that of its surroundings. As the Empire's capital, its location in the heart of the Iberian Peninsula prevented it from adding to its political and administrative function the varnish of a large commercial emporium".

A similar perspective has been defended more recently by Bel [24, 25] and Albalate y Bel [26], in concurrence with work by Madrazo [27] who, having rigorously analyzed the road network in Spain from 1750 to 1850, concluded that it had not been radial prior to 1700. "The network of roads of the Austrias had its origins in the network of Roman roads of the third and fourth centuries A.D., and had basic characteristics of mesh connections, forming a decentralized network, whose density was much lower in the northwest of the Peninsula (Galicia and Asturias) and in the southwest part of the plateau" (Bel, 2011, pp. 690–691 [25]). "It was not necessary to pass through Madrid to go from Burgos to Granada, from Oviedo to Seville, from Cartagena to Santiago de Compostela or from Huelva to Girona, nor was this necessary to go from Toledo to Valladolid, a route which could be undertaken almost in a straight line through Segovia or Avila. The same can be said for horizontal routes"(Madrazo, 1984, p. 152 [28]). In the same way, Uriol [29] concluded that the network was characterized by east-west itineraries

that ran through the great rivers' valleys, the north-south itineraries linked to these, and some diagonal itineraries connected with the former routes.

However, as mentioned early, neither Ringrose [30] nor Madrazo [27] directly analyzed Madrid's location advantages as a capital compared to other possible alternative cities. Ringrose was primarily interested in Madrid's adverse economic effects on its hinterland. These mainly affected Toledo, an industrial city confronted with Madrid's rapid urban development, which raised prices of food and clothes, reducing the competitiveness of its producers [9, 30]. Madrazo [28] attempted to identify the infrastructure work carried out under the Bourbons. One of his conclusions was that Madrid did not enjoy a central position in the road network until the *ex-novo* construction of the Bourbon radial roads. The upshot is that these influential works propagated the idea that Madrid's choice as the capital had no solid economic basis among Spanish historians [11].

García Delgado (1990, pp. 224–225 [20]) eloquently summarized this widespread belief:

*". . .the combination of its geographical situation, status as a city and centralized transport and communication network, offset the unfavorable conditions which Madrid presented for economic activity due to its poor natural environment; the scarcity of minerals, vegetation, and fuel as primary sources (except for some useful building material); its large distance from the most dynamic coastal trading centers and particularly the lack of fluvial traffic and inaccessibility to maritime transport, again due to distance from the coast.".*

Nevertheless, such a description of Madrid's disadvantages has several significant weaknesses. On the one hand, no city has all the resources it needs, but this problem can be circumvented if it can be easily provided for elsewhere. Linked to this, it is quite usual in many historical works on the subject [31–33] to underestimate the advantages of inland cities considering ship transport as much better than land transportation [34, 35]. On the other hand, the logistical benefits of cities from a network perspective are not considered since this would require a reliable reconstruction of the existing road network in the 16th century, something not achieved to date.

This paper revisits the debated economic suitability of Madrid as the capital of Spain from two perspectives not considered so far: that of the interactions between cities and that of the limited geographical knowledge of the time. We have not attempted to describe Philip II's personal decision-making process, but rather the rationale that may have been followed by anyone with similar objectives and knowledge.

The paper is structured in four sections. In the first, we analyze the economic suitability of cities from a traditional point of view considering only the characteristics that do not depend on the interactions between cities: population size, health, urban planning, and defense. To deal accurately with the other two major issues dealt with in the literature on the subject, namely the problems of supply and the impact of the capital city on the territory, it is necessary to consider logistical aspects that require a prior review of the existing transport network. Therefore, the following section is devoted to a thorough revision of the traditional vision of the Spanish transport network, contributing new primary sources to the debate and a methodology that allows us to reconstruct the road network in a map with a degree of detail that has not existed until now.

The Europe of the 16th century was marked by significant changes in cartography [36]. To avoid the common fallacy of judging the decisions made by Philip II with current geographical knowledge, we have elaborated a naïve version of the previous transport map. This naïve map, constructed with the planimetry of the 16th century, allows us to determine whether the

decision to locate the capital could have been influenced by a deficient knowledge on the part of the monarch of the geographical reality.

Both maps allow the application of the complex network approach which offers a new vision of the relationships between cities and the territory that is very different from those considered so far.

Using complex networks, the third section analyzes the advantages of the cities from the point of view of their commercial importance, their accessibility to the territory and the impact they would generate if they were to become a capital city. Complex networks is a relatively new area of scientific research, starting at the beginning of this century [37, 38], and only in recent years has it begun to be used in the field of historical analysis [39, 40]. It is characterized by the use of mathematical principles to assess the importance of the elements not in isolation but considering the role they play in the system.

Finally, in the last section, we present our conclusions and the possible implications of the results obtained for the purpose of future work.

## Advantages and disadvantages of candidate cities considered on their own

Many factors influence the suitability of a city to become a capital city. Given that in the 16th century the choice of the king had an important personal component, being well positioned in each of the characteristics considered did not mean being chosen but only not being discarded from the list. These are, therefore, aspects, which must be reasonably covered by the cities to be considered rational choices from an economic point of view.

### Population

It seems clear that the choice of the capital would fall upon one of the major towns to the extent that the size of their population represents an approximation of its economic, political, and cultural importance. The founding of capitals ex-novo has only been common in expanding territories since the costs and construction time are much higher.

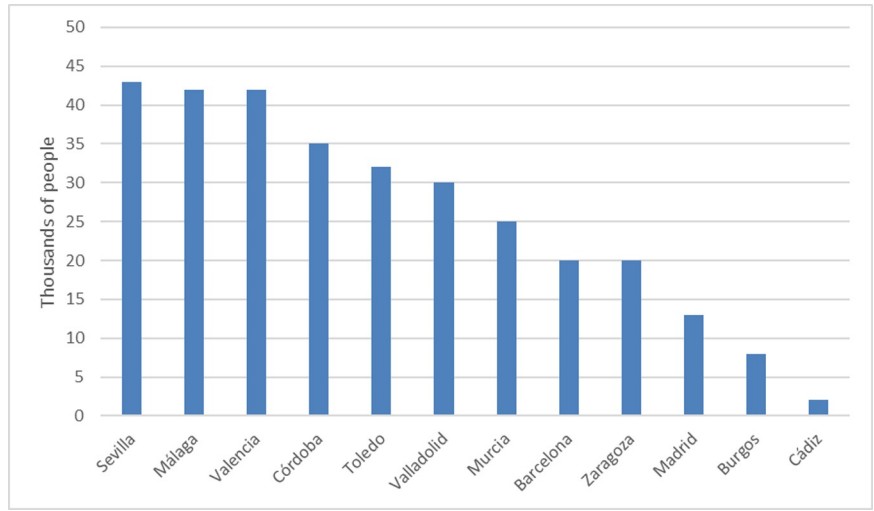

**Fig 1. Population of the main Spanish cities at the beginning of the 16th century.** Source: Bairoch et al. [41] and own elaboration.

According to data from Bairoch et al. [41], at the beginning of the 16th century, the most populated cities in Spain were in the south (Seville and Malaga) and the east (Valencia), all three having more than 40,000 inhabitants. Barcelona and Madrid were considerably behind this with 20,000 and 13,000 inhabitants respectively (Fig 1).

Thus, the size of the population does not seem too have been an important criterion of choice. However, the mere fact of being the capital generates a rapid increase in population, acting as a 'second-nature advantage from the perspective of Krugman [42]. So, even though initially not that big, the city soon becomes important, provided it has sufficient conditions to sustain it.

The influence of the capital on the population can be seen through the natural experiment carried out by Philip III, which involved the transfer of the capital to Valladolid in 1601 and its return to Madrid in 1606. The establishment of the court produced spectacular demographic growth in Valladolid. In just two years, its population almost doubled, from 36,000 inhabitants to over 60,000 [43], while Madrid suffered a sharp drop from 95,000 to some 65,000 inhabitants. The subsequent reinstatement of the capital produced a rapid recovery of the population of Madrid [44] and a corresponding fall in that of Valladolid. We can thus quantify the effect of 'being the capital' at around 30,000 people, a not insignificant figure considering that few cities reached that population at the time.

## Urban planning

The fact that Madrid was chosen despite its size indicates two things. On the one hand, Madrid qualified as the capital even with a population of a minimum size. On the other hand and given the difference concerning the previous and far more urbanized capitals, Valladolid and Toledo, this smaller population implied that the necessary urban planning of Madrid could be done to the taste of the king.

From the point of view of urban functionality, the narrow streets of Toledo, still organized along medieval lines, did not allow the court to move with adequate splendor and ease, making it difficult to ride on horseback through many of the streets and preventing movement by carriage along most of them [45]. Neither could Toledo accommodate the growing bureaucracy. It did not have enough houses, and its location on a hillock and surrounded by the River Tajo made it difficult to construct more [45].

Philip II, fond of architecture and urbanism as he clearly demonstrated with the meticulous supervision of El Escorial, was familiar with the urban developments of Flanders and the main Italian cities. Therefore, as Santos Vaquero [15] points out, it is likely that sinuous and consolidated urban layouts such as Toledo's were obstacles to the royal plans.

## Healthiness

The inherent "healthiness" of the location was also an essential factor. Prescott [46] considered that the main reason for Charles I's recommendation to his son to settle in Madrid was its healthy air.

Although it may seem strange to us now, the exposure to epidemics was a question of crucial importance, particularly when discussing the capital's location where the high number of travelers passing through proved an element of added risk. For example, the plague which struck Barcelona in 1589 produced more than 10,000 deaths, which resulted in the disappearance of a quarter of the total population [47].

The entire population, especially the wealthier social strata, was sensitive to this issue of health [48], so it is not surprising that the choice of the capital was influenced by it. Frequent

plagues caused the authorities, like the rest of the inhabitants, to leave cities in order to avoid contagion [48, 49].

Fig 2 shows the cities most affected by epidemics between 1500 and 1560 [48]. It can be observed how the cities most prone to the outbreak of disease were those of the Mediterranean coast and Andalusia. So, the advantage of long-distance communication offered by port cities was offset by the ease with which epidemics could spread to them.

In general, Castile was considered less exposed to epidemics, and it was believed that Madrid, in particular, had particularly 'healthy air'. Queen Isabel left Toledo to go to Madrid to recover from her illness in 1560, finding it much more pleasant, and it is believed that this could also have influenced her husband's decision [45]. Madrid also had an advantage over Valladolid and Toledo because of the difficulty these had in providing good quality drinking water. There were problems with the quality of water in Valladolid [50]. Toledo was the most problematic in terms of its water supply because of its high altitude and reliance on the River Tajo. The potable water consumption depended on cisterns that collected either rainwater or supplies from relatively distant springs. For other than drinking purposes, water had to be brought up the steep climb from the river by water carriers and machines [51, 52]. Only in the case of Madrid was there an abundant and accessible supply of drinking water of good quality.

## Defense

Although the Spanish empire was the leading military power during the 16[th] century, it was not free from attacks on its closest borders, meaning that the defense of the future capital was a crucial factor [45].

Furthermore, in the east, the towns of the Mediterranean coast were the scene of continuous Berber incursions [53]. Until the battle of Lepanto in 1571, they were also vulnerable to Ottoman attack [54]. The frequency of the corsair attacks motivated Philip II to expound on the necessity of fortifying the coast before the Aragonese courts in Monzón [55, 56]. These attacks were not only limited to the coast but reached up to 50 kilometers inland, taking advantage of the limited navigability of the Ebro, Jucar, and Segura rivers [28].

> [Referred to 1509] *"Things were different in the water: because so many corsairs left Africa that you could not sail on the coasts of Spain"* [57].

The north coast was not free from risk either. In 1589, the Drake-Norreys expedition aimed to take advantage of the failure of the Spanish Armada in the previous year to attack the Cantabrian ports and destroy what was left of it. They wanted to take Lisbon, crown Antonio de Crato as King of Portugal and thereby establish a permanent base in the Atlantic by conquering the Azores, facilitating British attacks of convoys from the Americas [58]. During the second half of the 16th century, the owners of La Rochelle harassed the coasts of Asturias and Vizcaya in the broader context of the religious wars, looting villages and capturing merchant ships, even after the Peace of Vervins in 1598 [59].

The border with France was also a sensitive area. Even the signing of the peace of Cateau-Cambrésis in 1559 did not diminish the tension between the two kingdoms, as evidenced by the strategy of fortification of the frontier during the second half of the 16th century [60].

Finally, in the south, the defense of the coast was not assured either. Cadiz, for example, would be taken and plundered by English troops in 1596. Seville was the port city with the most significant defensive advantages of the Andalusian cities. The navigability of the Guadalquivir could potentially become a defensive problem, allowing enemy troops to land upstream, as occurred in 844 and 859 [61], although the availability of artillery in the 16th century almost

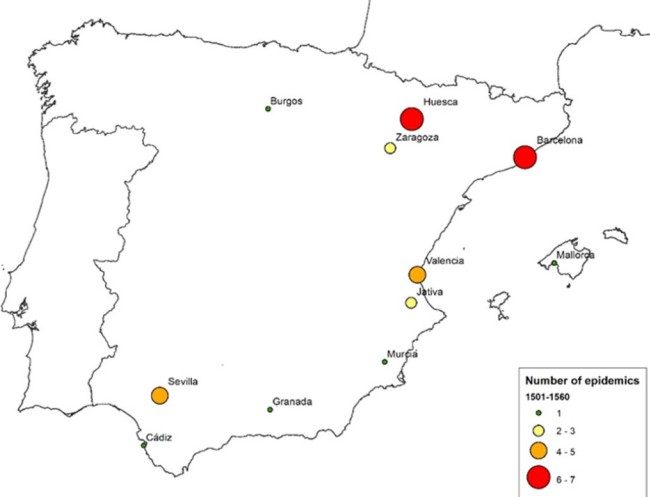

**Fig 2. Spanish cities most affected by epidemics during the 16ᵗʰ century (1501–1560).** Source: Villalba [48] and own elaboration. Base map from the Centro Nacional de Información Geográfica (Spain).

eliminated this danger. Andalusia also suffered the disadvantage of the tense situation with the Moriscos, as demonstrated by the rebellion of the Alpujarras between 1568 and 1571, a significant cause of concern for Philip II.

All the maritime capitals of Europe are in easily defendable estuaries [62]. The situation of Valencia and Barcelona was quite different. Barcelona's defensive problems are evidenced by the more than fifteen times that it has been besieged and attacked from the sea. Valencia's defense was not much better. Although Valencia was situated several kilometers inland at the time, its vast beaches facilitated easy landings, making it vulnerable to attacks too. The safest ports, Santander and Cartagena, were not considered potential capitals due to their transport difficulties by land.

All this suggests that, from a defensive point of view, the locations in the interior of Castile, such as Valladolid, Toledo, or Madrid, would be considered more convenient as compared to coastal areas like Barcelona or Valencia. According to Fernández Álvarez [17], there was only one safe land par excellence, the Castilian plateau, which was comparable to an impressive castle surrounded by high mountains, a circumstance that must have been considered as one that conditioned the decision of Philip II.

Two important issues of an eminently economic nature have not been still mentioned: the possible problems of supply of food and other goods for a city the size of a capital and the effect this could potentially have on the development of the surrounding territory.

These aspects will be dealt with in detail later, but for this it is necessary to analyse the transport network of the time, as they are critically dependent on it.

## The transportation network in the Spain of Philip II

In general, from the point of view of communications, defense and commerce, a central location was considered the best option for a royal residence, as recommended by the most famous and renowned treaties of the Renaissance [45]. Along with defense and health, the ease of supplying goods and provisions, along with all logistical matters, became necessary.

However, these advantages of centrality are not intrinsic to a specific geographical position. Moreover, they depend critically on the territorial distribution of the villages and towns and the road network's configuration. Hence even a city in a considerably unusual position may

offer better logistical advantages, i.e., maximum centrality, as determined by its position with respect to other cities and existing communications network. Conversely, the center will only have a full logistical advantage if the distributions of both the population and road are homogeneous. Therefore, a town's centrality cannot be considered autonomously but must be defined in tandem with its connectivity with other towns and cities. Thus, the accurate evaluation of the transport network is critical to determining the best location of the capital.

## The traditional vision of the Spanish transportation network

In general, practically all the work on the logistical aspects of the choice of capital and 16[th] century land transport adopts the road network which appears in the itineraries of Villuga [63] and Meneses [64] as a reference, completing it, in the best of cases, with the routes appearing in the travelers' stories of the time [18, 24, 45, 65–70].

The picture obtained is a decentralized network formed by homogeneous roads of low quality. The highest road density was found in the area demarcated by the main Castilian cities. On the other hand, there are areas with very few or even no roads on the Cantabrian coast, eastern New Castile, Extremadura, or southern Aragon. Finally, the network offered low connectivity of inland areas with the coast and a notable lack of coastal connectivity except in the cases of Catalonia and Valencia.

Based on this picture, the area with the best conditions for establishing the capital was located around the Valladolid-Madrid-Toledo axis. However, none of these three cities stood out as having more significant advantages than the others. At the same time, Seville's position was considerably worse due to the difficulty of access to the Spanish plateau. Using network analysis and GIS, Carreras Monfort and Soto [70] (63) corroborate this interpretation (Fig 3).

In the historical analysis, the different cities' logistical advantages are usually measured by the transport costs per ton. However, what really matters is the unitary transport cost, i.e., the proportion of the final price of supplies due to transportation, which is high when long-distance is combined with a low value-to-weight ratio. This can considerably reduce average transport costs since most cities' supplies, mainly food and low-price manufactured products, came from their surroundings. Bread alone could constitute up to 40% of the supplies to an urban nucleus [71].

Estimates for Madrid of the distances from which products such as bread or wine were brought are between 100 and 200 kilometers [72, 73]. Other products, such as meat or fish, came from even greater distances [14, 74, 75], so that transport costs could considerably affect the prices of certain products. The available evidence indicates a slight variability in the prices of products between different cities, which implies an appreciable degree of market integration [76, 77] supported by a transport network that worked quite well within the parameters of the time [8].

As was the case of the rest of 16[th] century Europe, the Spanish road network consisted of low-quality routes, along which transport was often easier on the backs of beasts than on wheels. Occasionally, the road conditions were so bad that it was preferable to travel cross-country [45].

The low quality of the roads and the country's rugged geography made land transport slow and expensive. Added to these problems was the scarcity and low quality of inns and the danger of banditry, especially in Catalonia. In 1565, Philip II considered that in the Principality of Catalonia, "It was impossible to travel . . . without clear risk to life" [27].

These issues meant that the cities near the sea or a navigable river enjoyed significant logistical advantages since mules and carts were significantly slower and more expensive than ships.

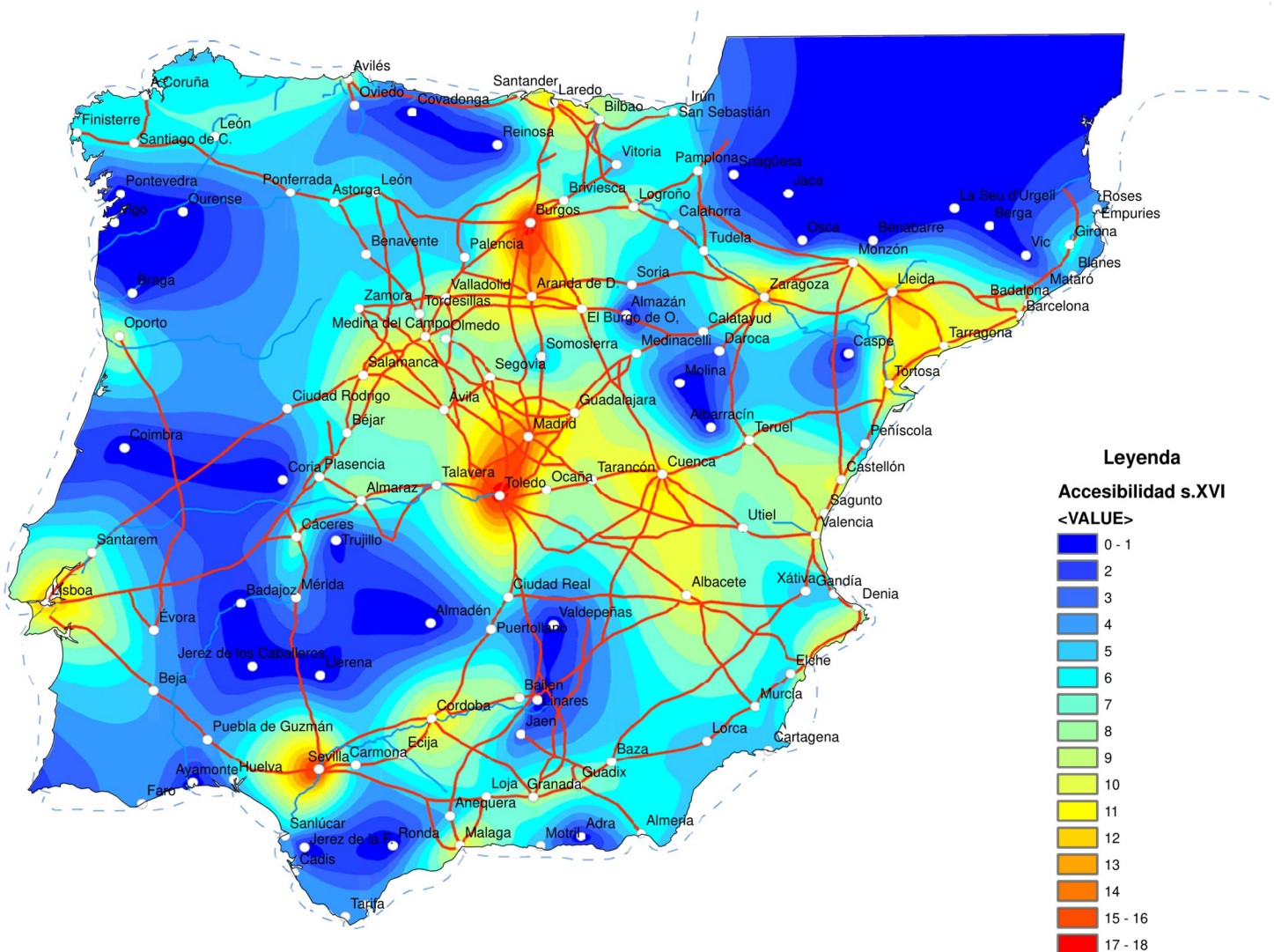

**Fig 3. Accessibility in the 16<sup>th</sup> century.** Source: Reprinted from Carreras Monfort and Soto (2010) under a CC BY license, with permission from UPC University Press, original copyright 2010.

Historical analysis combining geographical information systems (GIS) [13, 78] and spatial networks [79], has improved our knowledge of journey times and the intermodal transport costs of goods and people [39, 70, 80, 81]. Carreras Monfort and Soto [70] estimate that land transport could be forty times more expensive than fluvial and up to fifty times more expensive than maritime transport.

The comparison of transport costs for Madrid and Barcelona indicates the logistical advantages of the coastal cities compared to those in the interior (Fig 4).

### The overvaluation of transport by ship in terms of city supplies

These results, however, seem to severely overestimate the advantages of coastal cities by focusing on the most favorable conditions for transport by ship whilst forgetting some of its main weaknesses. As Ballaux and Blondé point out [35], the lower costs of ship transport did not

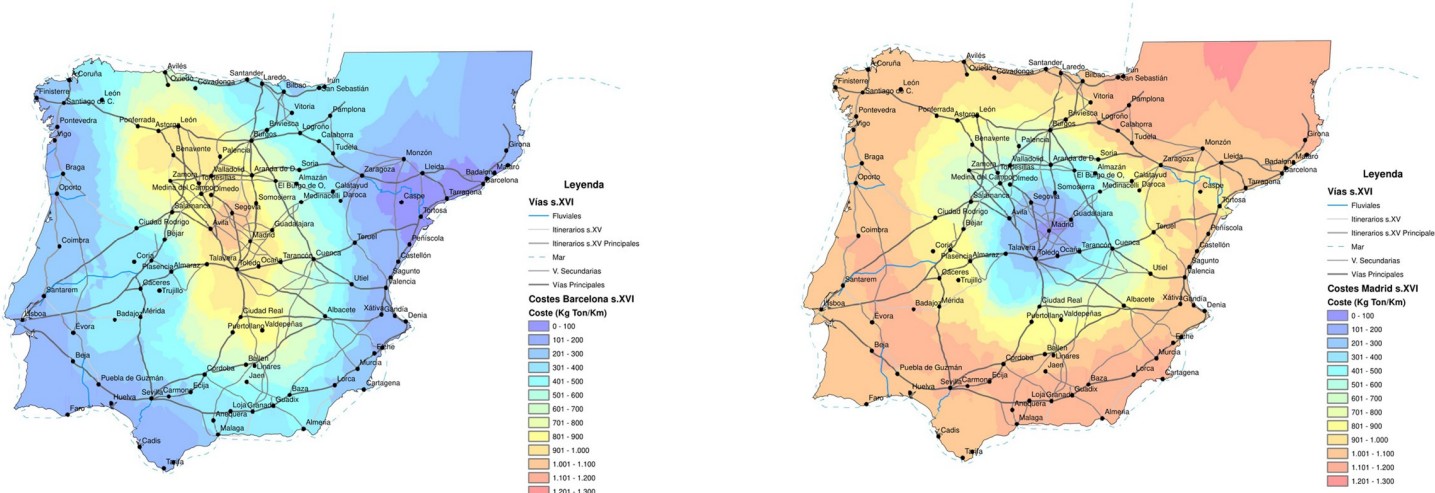

**Fig 4. Comparison of transport costs for Barcelona and Madrid.** Source: Reprinted from Carreras Monfort and Soto [70] under a CCBY license, with permission from UPC University Press, original copyright 2010. A. Barcelona, B. Madrid.

prevent a significant share of long-distance international trade in the 16th century from being carried out by land rather than by maritime means of transportation.

Although a port's availability is a positive aspect for the supply of goods to a city, it is not decisive since ship transport also implies some negative factors.

1. As abovementioned, what appears relevant, from the point of view of the adequacy of the location of the capital, are total transport costs. These not only depend on the cost per ton per kilometer but also on the origin and nature of the products transported. Being located by the sea can significantly increase the transport costs of products from the hinterland since it is necessary to travel long distances in order to achieve the same supply area. For an average coastal city with no convexity or concavity to its coastline, the distance to the edge of its hinterland is more than 40% greater than that for an inland town. Hence, the most significant logistical advantages are for cities with navigable rivers, but which are located not far from the coast such as Seville, Hamburg or London. For a coastal town to be attractive as the site for a capital city, this disadvantage in terms of provision needs to be compensated for by the logistical advantages granted by its port and foreland. This idea of the adverse effects of the lack of a piece of territory on economic growth was contrasted by Redding and Sturm [82] using Germany's division and subsequent reunification after World War II.

Therefore, the supply advantages of the coastal cities were not generalized but applicable only to certain products. The towns of the interior, particularly Madrid, had an advantage, for example, in terms of wheat supply [83, 84], although the variability of prices was higher than in coastal cities since they lacked the stabilizing effect of imports in times of scarcity [77].As most of the population's supplies were food products from each city's relatively nearby area, the average transport cost per unit of supply was relatively low. For example, those that did come from far afield, such as products demanded by the aristocracy originating from abroad, were a small percentage of the total and were of a high value to weight ratio, reducing the relevance of transport costs. The port cities' logistical advantages depend on the network's latency and transported goods. This network latency, produced by the waiting times necessary to continue the trip, is a crucial aspect rarely addressed in historical transport studies. These delays, studied by the theory of temporal networks [85], affect the traveling time and the costs since

they imply the need for storage and preservation in perishable products. Although the latency of the network arises with all means of transport, it is especially high in transport by ship.The low traffic volume of goods and the high latency of the maritime network in the 16[th] century can be illustrated by the small number of ships arriving in Valencia. According to data from the Sea Toll, it was about 400 or 450 per year, just one or two a day and usually tiny ships. If we look at the traffic from Barcelona, the most important origin of medium-distance cabotage traffic, only about nine ships per year arrived on average, i.e., less than once a month [86]. This flow represented a small percentage of the total since most of the arrivals were small vessels [87]—from other ports along the Valencian coast such as Denia, Vinaroz, or Alicante.

This high latency was a negative aspect of coastal shipping compared to road transport since it forced merchants to wait until there was a ship with the desired destination. This circumstance could imply a very remarkable lengthening of the journey time, especially if, as a feature of the 16th century, the variability of ship arrivals was very high due to storms or pirate attacks. The application of queuing theory [88] shows that the average expected waiting time for the departure of a ship could significantly exceed the average waiting time.

A simulation exercise was carried out to see the impact of this variability on the average waiting time. To perform the simulation, the annual series was first completed by linear interpolation, and then a thousand simulations were carried out assuming that the arrivals followed a uniform random distribution throughout the year. This is a lower limit of the expected waiting time since it only considers interannual variability. Seasonal variations could increase waiting times. The calculations were carried out, always considering the waiting experience of the previous two years.

The results obtained show that it was necessary to wait for at least 20 days for a ship from Barcelona but that it was not uncommon, at certain times, to have to wait for 100 days or even more, depending on the number of ships arriving at the port each year (Fig 5).

Consider that according to Carreras Monfort and Soto [70] for the 16th century, traveling from any point of the plateau to the periphery, which is approximately the distance between Barcelona and Valencia, took about 15 days. Compared to road transport, the advantages of maritime transport practically disappear in terms of speed when the latency is included.

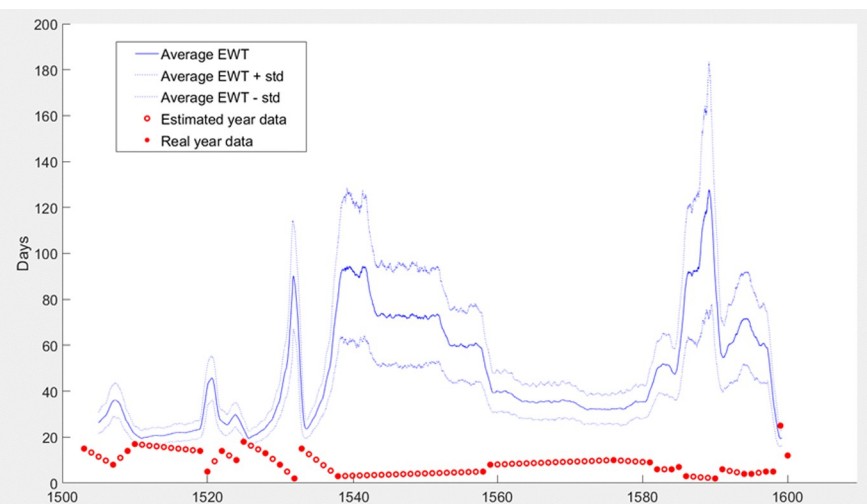

**Fig 5. Expected Waiting Time (EWT) for arrivals of ships from Barcelona in the port of Valencia (days).** Source: Salvador Esteban [86] and own elaboration.

The high latency resulting from fleet limitations in terms of capacity and vessel numbers was frequently increased by the bleak weather and corsair attacks which further reduced the initial advantages of navigation over travel by land. The benefits of overland transport for essential and urgent goods are evidenced by the fact that the couriers established land routes in preference to sea routes [89]. The danger of piracy affected even the cabotage traffic. It was not uncommon to wait for an accompanying squadron to make the trip or opt for an alternative journey by land [28].River transport was widely used in 16[th] century Europe and shared much of the cost advantages of maritime navigation, but with lower risks and lower latency. In general, there is an inverse relationship between vehicle capacity and latency. Oversized vehicles are more efficient and have fewer costs by transported ton, but they are less flexible and cannot always be used.

Although large rivers are excellent communication channels, they are also important obstacles [90]. They favor transport when the itineraries are parallel to their course, but they hinder or prevent it when they run perpendicularly. This means that the benefits generated by river transport are not widely spread throughout the territory if the rivers or channels are not integrated into a network. The advantages of fluvial over terrestrial transport come to the fore when the trajectories are parallel. However, this is not usual, and the existence of meanders, or the channel's inadequate orientation concerning the intended destination, can significantly reduce these advantages. Thus, for example, the length of the Ebro River is 930 kilometers, but the equivalent distance by land is 602 kilometers. To make the most of the river transport goods between Zaragoza and a city relatively close to the river, such as Lerida, at 142 kilometers, it would be necessary to travel 221 kilometers by boat to Mequinenza, then 42 kilometers by land. To save 100 kilometers of the land journey, we would have to travel 221 kilometers by boat and add the cost of internodality. The fully navigable Zaragoza-Barcelona itinerary would require a 509 kilometers journey by boat compared to 295 kilometers on land.

A natural experiment regarding the advantages of fluvial transport versus land transport is the Canal de Amposta. This channel, whose stated objective was to facilitate Aragon's trade with the Americas, was begun in 1778 and would have made the Ebro navigable up to some 450 kilometers by saving the problems of navigability of the Delta. However, the project was abandoned five years later when it became clear that the high maintenance costs of just 10 kilometers of the canal did not compensate for the possible savings in transport costs [91].

In the Iberian Peninsula, there are few navigable rivers, unlike other areas of Europe. Only Seville enjoyed natural navigability of some depth. However, this river port's advantages were limited in the years concerned due to an absence of a network of river tributaries for distribution and poor road communication with the rest of the peninsula.

In selecting the capital, the disadvantages of cities that did not have a river port could be compensated if it was feasible to connect them through canals. In the Europe of the early modern era, the construction of canals for navigation and irrigation was not unusual. In Spain, from 1524 to 1681, several studies were carried out to evaluate the possibilities of navigating the Guadalquivir, Pisuerga, and Tajo rivers [92–94]. Although nothing very significant came out of the possibilities identified until two centuries later, with the Canal de Castilla and, to a lesser extent, with the Canal Imperial de Aragon, it can be speculated that, at the moment of his decision concerning the capital of Spain, Philip II would have considered Valladolid, Madrid, and Toledo as potentially navigable. This idea is by no means far-fetched when we think that the project was still considered viable at the beginning of the 19[th] century, almost three centuries later [95]. This only changed when the railway's arrival finally eclipsed the canal as an economical mode of transport.

Cost differences between modes of transport should be reflected in freight flows. Cheaper and faster modes should show a higher share of traffic. Munro [34] argues that in 16[th] century

Europe, land transport showed significant advantages over sea transport as it channeled most of the traffic between Italy and the Low Countries.

The information available for Spain only allows us to calculate the traffic generated by goods unloaded in some ports. Nevertheless, it can be affirmed that maritime traffic was minimal compared to land traffic.

Based on the estimates of Madrazo [27], land transport capacity in the 16<sup>th</sup> and 17<sup>th</sup> centuries is estimated at 300 million ton-kilometers per year. With these data, the total trade from the Americas would only occupy at most 5.75% of the annual capacity if the average distance traveled by the cargo had been 250 kilometers, being reduced to 2.4% if the distance was 100 kilometers, which is more plausible [96].

Regarding the northern ports, the lack of data forces us to assume that the number of ships bringing imported goods coincided approximately with the number of ships leaving with exports. The number of ships leaving the northern ports [97, 98] loaded with wool [99] and iron [100] would be around 65 ships, which gives us similar results to those of the trade with the Americas. The information for the Mediterranean ports is even more limited except for Valencia. Still, a rough estimate would give about 40,000 tons, including a part of cabotage, which would mean an annual land transport capacity occupancy of between 2% and 4%.

In total, freight traffic with the ports, including loading and unloading, would account for a maximum of between 6% and 8% of annual land transport capacity. From the point of view of the impact on traffic in the cities, this is a relatively small effect, since it would have been distributed not only among the ports but also among all the towns affected by this traffic, including those in the hinterland.

To the extent that the relative costs of the different modes of transport determine trade flows, they also determine the road network's development. If the advantages of shipping transportation are much more significant than those of road transport, the network will be configured to take full advantage of them. Fig 6 illustrates this issue. If transportation by ship is more convenient than land transport to go from one city to another, ships will be used as much as possible, generating more and better quality roads linked to the coast. On the

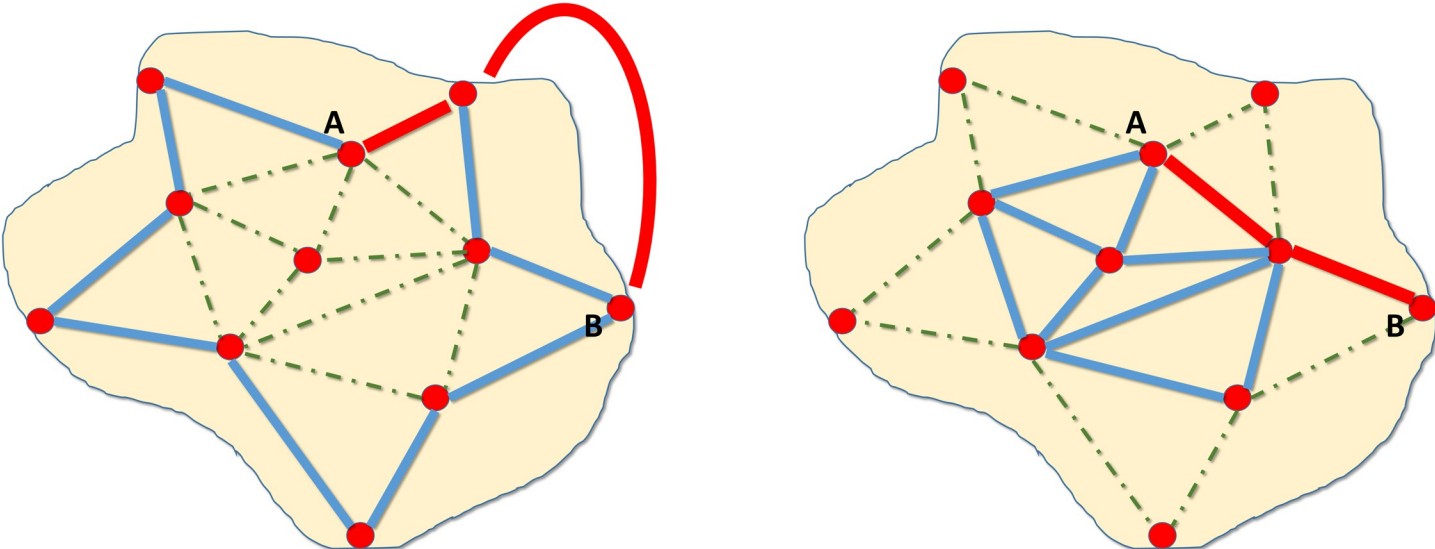

**Fig 6. Relative advantages of maritime transport over land transport and development of the road network.** Fig 6A) Road network with advantages in ship transportation. Fig 6B, Road network with advantages in road transportation. The blue lines represent the main roads, and the green dashed lines the secondary roads. The red lines show the optimal routes between A and B depending on the cost advantages of the means of transport. Source: Own elaboration.

contrary, if it turns out that land transport is better, most roads will be concentrated in the interior of the territory since there is the intermediation centrality and, therefore, traffic is more significant [79, 101].

The routes shown in the Villuga and Meneses itineraries are not consistent with significant advantages of maritime transport over land transport since the road connectivity of Spanish ports with the interior is reduced, and most of the essential routes are concentrated in the interior of the peninsula [67].

To determine more quantitatively the differences of costs between the two means of transport, the centrality of intermediation and the accessibility of the main cities were calculated using different hypotheses. They were subsequently compared with the map of roads most frequently described in Villuga's itinerary [18] and the road network of the 18th century [101] (Annex 1 in S1 Annex). The results show that maritime-land cost ratios below 1 generate traffic flows inconsistent with the observed road network. This implies that in 16th century Spain, land transport was in most cases at least as convenient as maritime transport, which is entirely consistent with the results obtained by Munro [34] and Ballaux and Blondé [35] for trade between Italy and the Low Countries.

We can intuit the effect that a river communication with the coast would have had on the supplies in Madrid, considering the connection by railroad with the port of Alicante in 1858. The data shown by Reher and Ballesteros [102] show that the railroad did not significantly alter the evolution of consumer product prices in Madrid. Likewise, they indicate that it was possible to supply the capital by road transport alone without substantial differences in prices with respect to coastal cities such as Barcelona. These results, referring mainly to the 19th century, are not due to the effect of market integration by the Bourbon road network since they were also produced beforehand [103].

In short, a location near the sea or a navigable river, while favorable, is not a decisive factor in the choice of location for the capital, at least in a territory with the orographic characteristics of the Iberian Peninsula. Proof of this is that from the 3rd century until the fall of the Western Roman Empire, Mérida was the capital of the Diocesis Hispaniarum. After the barbarian invasions, Toledo, another inland city, was the capital of the Visigothic kingdom until its conquest by the Arabs in 711.

During the Reconquest, the fragmented Christian kingdoms that emerged after the Arab invasion also tended to establish their capitals, not always clearly defined (Fig A7 in S1 Annex), in cities without communications by ship: Oviedo, León, Santiago, Burgos, Pamplona, Zaragoza and Toledo. Only Barcelona, and later Valencia and Lisbon were port cities.

In Al-Andalus, the capital was Cordoba from the beginning of the Caliphate until its fragmentation at the beginning of the 11th century. At the time of its greatest splendor Cordoba was the main city of the peninsula and one of the largest in the world, with nearly half a million inhabitants [41] which were adequately supplied even though navigation on the Guadalquivir was practically non-existent. Notably, Madrid did not reach that population size until the beginning of the 20th century. Nor did the capitals of the Taifa Kingdoms show any tendency to being located predominantly on the coast.

Therefore, it seems unlikely that the problems of supplying goods and food would be thought critical, especially if we consider that the choice of the capital was not seen at the outset as an irreversible or long-term decision. Anyway, inner cities such as Seville, Toledo, Valladolid, Madrid were good in terms of food supply [50, 72, 104], both because of the fertility of the lands which surrounded them and from a logistical perspective, although Seville had the important added advantage of the navigability of the Guadalquivir [105].

However, the supply of drinking water was of utmost importance. The location near a large river may seem to be a decisive advantage in this respect, but it was not always the best option

since it was necessary to elevate the water to the city, and it was not always free of sediments. Also, the water could become contaminated due to the inadequate management of sewerage at the time, which tended to lead to the outbreak of epidemics. Since Roman times, the availability of clean water from the mountains had been considered far more desirable, as can be seen by the many aqueducts built to carry water to populations along significant rivers [52]. The abundance of good quality water could have offset the higher cost of other supplies when considering the optimum location. In this respect, Madrid [6] had an advantageous position compared to other cities such as Seville [106], Toledo [15] or, Valladolid [50].

## The biased view of the road network

Previous evaluations of the most suitable locations for the capital settlement have focused on the cities' characteristics, without a detailed study of the topological relationships between them. The best place may be useless if most of the country's population is concentrated on the opposite side of the territory or if a mountain range seriously hinders commercial traffic from that city.

To take all these issues into account and in order to obtain an overall view, it is necessary to accurately represent the territory and the roads between the cities. However, sufficiently detailed road maps of Spain did not begin to emerge until the end of the 17th century. Faced with this situation, it is necessary to build the map of transport routes using the substantial but dispersed information available.

The problems involved in carrying out this work are important and very diverse. However, they fall into two main groups; also included relevant road information and lack of road categorization.

**Incompleteness.** Menéndez Pidal [18] constructed a road map for sixteenth-century Spain in accordance with the renowned itineraries by Villuga [63] and established a hierarchy of roads according to the number of times that the itineraries were described. This methodology is applied in Fig 7.

We have started from these same itineraries and those of Meneses (57), which are very similar. However, we have added information to them from other lesser-known sources; D'Ocampo [107], Estienne [108], Gail [109], Stella and L'Herba [110], Rowlands [111], and the exceptional road map of Hogenberg [112]. We do not include information before the 16th century, such as Scriptor [113] or Itinearium de Brugis [114], or incomplete information concerning the Iberian Peninsula, such as Etzlaub [115] and Walseemüller [116].

However, the combined information from all the sources shows some important towns being cut off. It would not be reasonable to accept this, as multiple sources indicate that it was possible to access almost all the towns of the time on horseback or even by cart.

Let us compare the Villuga [63] and Meneses [64] itineraries with a map of the time, such as that of Forlani Veronese [117]. We observe that many important towns do not show any interconnection with the neighboring population settlements, a sign that not all of the roads are included, but only a selection. In particular, the zones with more detailed information correspond primarily with the courts' headquarters and places of pilgrimage.

It could be argued that some roads do not appear because they were low quality, barely passable paths for which the traffic of people and goods was challenging.

The verification of this hypothesis is complicated because the evolution of the network has meant that there are hardly any remains that allow a generalized determination of the quality of the roads in the 16th century. However, its transport capacity can be checked indirectly by using geological analysis. Minerals differ depending on the quarry from where they were extracted. Therefore, if we analyze the materials used in construction, we can make a guess as

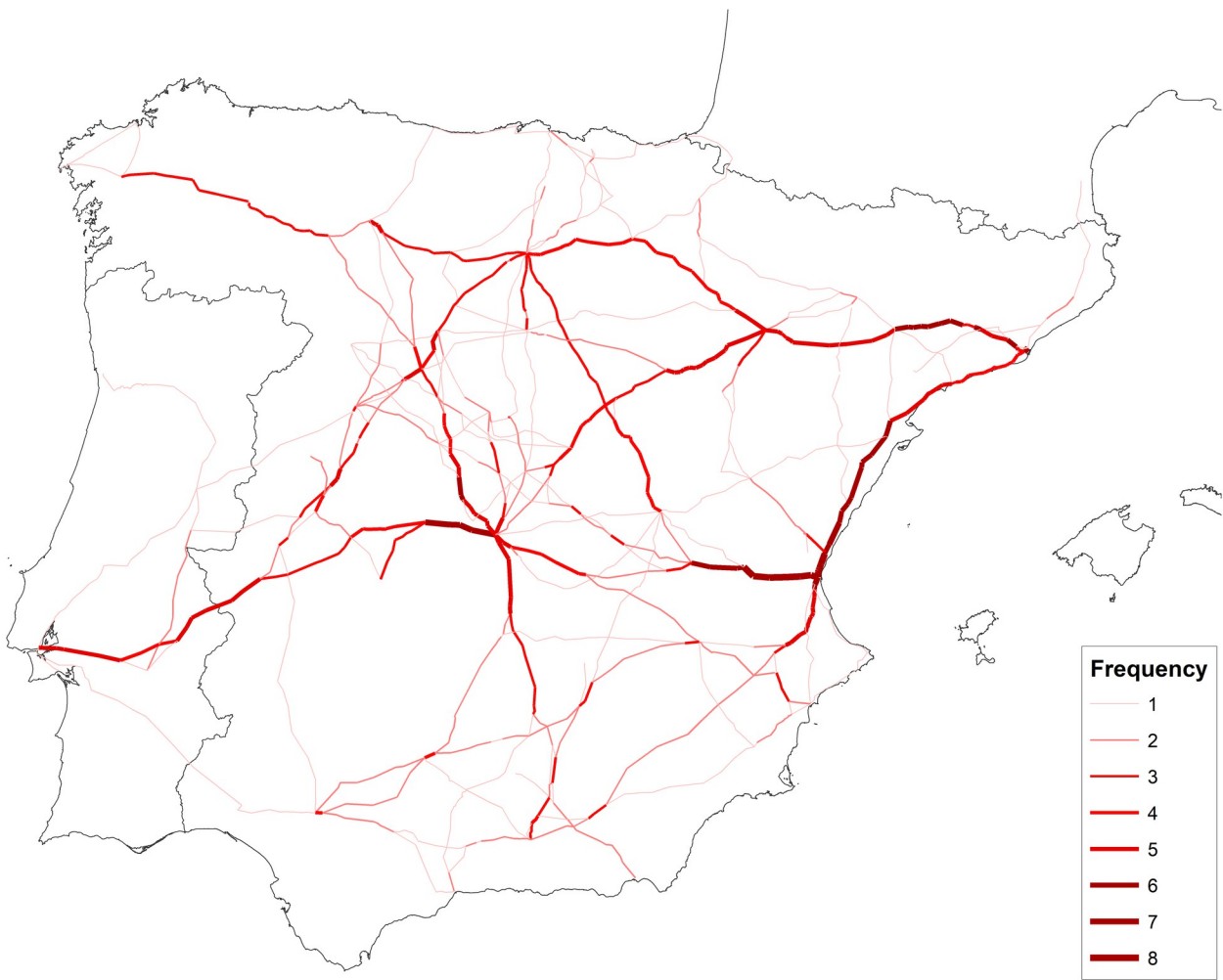

**Fig 7. Roads most frequently described in Villuga's itinerary.** Notes: Villuga's itinerary does not offer an explicit classification of the importance of roads, but the frequency with which the road sections appear can serve as an approximation. *Source*: Villuga [63] and own elaboration. Base map from the Centro Nacional de Información Geográfica (Spain).

to where they were transported. The use of masonry stones requires transport by a cart, which implies roads of a certain quality. The Villuga and Meneses itineraries [63, 64] do not show the roads used to transport the vaugnerite and leucogranite rocks used to construct the Renaissance monuments of Salamanca. Their quarries were located in Ledesma [118] and Martinamor [119].

Likewise, the intense trade of heavy millstones between distant towns using roads that do not appear in the itineraries reinforces this hypothesis [120, 121].

The map titled "Spagna con le distantie de li loci" [122] shows a communication network quite different from the one that appears in the itineraries, which indicates that many of the roads that linked important towns are not included in them and, therefore, that the road network was much denser than is usually considered. However, the fact that essential roads do not appear on the map is an indication that the quality of the roads was similar (Annex 2 in S1 Annex).

Finally, the road maps of the early 18[th] century show a very dense network. Given the scarce investments made in the two previous centuries, we can assume it would not make a big

difference compared to those existing two centuries earlier. The minor differences between the 16[th] century itineraries of Villuga and Meneses [63, 64] with those of Pontón of 1705 [123], corroborate the hypothesis of these scarce improvements.

These are just a few examples, but they illustrate the omission of essential roads in the Villuga and Meneses itineraries [63, 64].

The selection of a small number of roads may be due to the application of stringent road quality criteria but without altering the appearance of the basic structure of the network. Posing a more serious problem is the possibility that the selection process was biased, and that the road density of some areas does not correspond to reality. Did the 'gaps' mentioned by Madrazo [28] correspond to reality, or are they simply due to insufficient information collection or to a lack of interest in those areas?

The available historical information indicates that the road density was much more homogeneous than is usually thought.

One of the most notable 'gaps' is the Pyrenean area. However, the map drawn by Vrints in 1608 gives us a completely different image for the Catalan part [124]. Despite not explicitly including the roads, we observe that the communications in the north of Catalonia were of good quality through the distribution of the bridges. We can see this even more clearly in Sanson's map of 1652 [125], in which seven roads crossing the Navarrese Pyrenees can be appreciated, while in Villuga and Meneses [63, 64], only one appears. The real network seems to have been even denser; the improvements in cartography enabled Sanson, three decades later, to make a much more detailed map of the whole of the Pyrenees in which 34 ways are shown, all of them categorized in the same manner [126].

Galicia, another of the significant 'gaps,' was not only connected with the rest of the peninsula and Europe by the French Pilgrim Route, as it appears in Villuga and Meneses [63, 64]. The Portuguese, The Northern, The Primitive, and The Sanabria routes also reached Santiago. All of them were significant roads that made Santiago one of the leading European pilgrimage destinations.

If we invert the argument used by Madrazo [28] and Uriol [123] that little investment was made in road infrastructure during the 16[th] and 17[th] centuries and that, therefore, the network of the early 18[th] century was very similar to the one reflected by Villuga and Meneses, we can also use the maps of the War of Succession compiled by Wit [127], Fer and Loon [128], Valk [129] and Visscher [130]–in order to glimpse the type of network that existed at the time the capital was chosen.

The image that these maps portray is a very homogeneous network both in its distribution and quality. There are no differences in quality between some roads and others. So, given that they were created for the movement of troops, we can assume that they were all acceptably functional, even for carts and artillery trains.

If we consider the large 'gap' in the southwestern area on the 18[th] century maps, we can see that, far from the absence of roads, there was a relatively structured network between the main towns in the area. It is possible to observe the critical road from Almadén to Seville essential for the transport by cart of the mercury needed for the precious metal mines of Peru and Mexico, the existence of which has been documented since at least 1558 [131].

Although there was a vast network across most of the territory with many alternative roads, accidents of orography meant that different traffic flows frequently had to pass through the same point. This naturally gave rise to important commercial opportunities, which led to conflicts, even violent ones, between neighboring regions to control these passages. Or, at times, these communities would prevent the construction of alternative roads [132].

The alleged lack of connectivity of the coast 'which forced cabotage, unless one came inland more than a hundred kilometers' [28] deserves special attention. The need for defense against

the Berber attacks had created a Mediterranean coast lined with watchtowers at short distances from each other [133]. It seems improbable that no roads connected them, since they would have proved necessary to allow garrisons to maintain and transport troops. The paths that Uriol [67] selects for the 15[th] century show coastal connectivity that, while not complete, is far greater than that shown in Villuga and Meneses [63, 64]. The cartographic sources of the early 18[th] century delve into this idea, indicating that the coastal towns were perfectly connected between each other and the interior, which seems logical given its economic and defensive importance. Finally, this hypothesis is supported by a source from the 16[th] century itself, little used but of great relevance, being a complete description of the coastal path around the peninsula that D'Ocampo [107] details in the first chapter of the *General Chronicle of Spain*.

A review of the municipal documents shows that few towns were not, at one time or another, involved in the repair of a bridge, the fixing of a wrong step, or the conservation of a stretch of road [27]. The fact that practically all the towns participated in the maintenance of the network is indicative of a dense network of national interest despite the absence of direct state intervention.

In general, we can say that the territory was crisscrossed by a vast road network that connected some villages with others through an intricate labyrinth of local and regional roads that forced the traveler to choose one or the other, depending on the circumstances. The routes—the imaginary lines that, in the words of Delano-Smith [134] link a starting point with a destination—were decided before starting the journey based on the information available in maps and itineraries. Nevertheless, once the journey began, the path was chosen at a moment's notice, often by asking the locals and other travelers.

As the road network cannot be reconstructed entirely from primary sources, some artificial generation is necessary. The most common approach to the problem is the application of mathematical procedures to generate links that meet certain conditions of proximity so that the resulting network coincides as much as possible with the available evidence.

Mathematical approximations usually have a local character, based on Delaunay's triangulations [135], with specific criteria being applied to reduce the number of links. β-skeletons are helpful because they allow the generation of efficient transport networks [136] with different road densities corresponding to different values of β. The network's density is directly related to the average degree centrality of the nodes [137].

The main problem with these local approaches is that they do not include path dependence and coordination between nodes in the development of infrastructures. Approaches based on network analysis [138] or agent-based simulation [139] present two main disadvantages. Firstly, they need assumptions to construct the models, and secondly, these approaches are computationally intensive.

In recent years there has been a remarkable development of bio-inspired approaches. The evolution of the Spanish road network has also been analyzed through slime mould mapping by Adamatzky and Alonso-Sanz [140]. However, this methodology suffers severe problems due to the high sensitivity of the results to the variations in the diffusion of the chemo-attractants [101].

The networks of Delaunay [135] are a good way to represent road communications because they generate structures similar to those observed in reality due to their efficiency properties [139, 141]. These networks provide accessibility such as that of complete networks with fewer roads. The use of a low number of roads to achieve communication between the populations not only reduces the costs of creating and maintaining infrastructures but also increases the traffic on the roads, with the subsequent increase in economic activity and safety.

Another property of the Delaunay networks is that they ensure that the maximum length of the itinerary between two points will never exceed 2.42 times their Euclidean separation. In a

scenario with high differentiation in road quality, such as today, travelers may choose far longer itineraries in return for cost savings. However, in the 16[th] century, the poor quality of roads made it unlikely that a detour was preferable to the direct route. Also, the low quality of the roads meant that construction costs were low so that if an existing road was too long, shortcuts were created easily, increasing the density of the network.

To verify that the Delaunay network was a good predictor of the road system, its results were compared to the itinerary information available in the historical sources (Fig 8). To ensure the comparability of the roads, it was necessary to previously determine the nodes which would be considered in the analysis and homogenize the links. One thousand and ninety-six nodes were selected, representing approximately 65% of the peninsular population of the kingdoms of Castile, Aragon, Navarre, and Portugal. These nodes comprise all the towns that appear on Forlani Veronese's map [117] and the main towns indicated in the itinerary sources. For technical reasons, some crossroads, which function as Steiner points, were

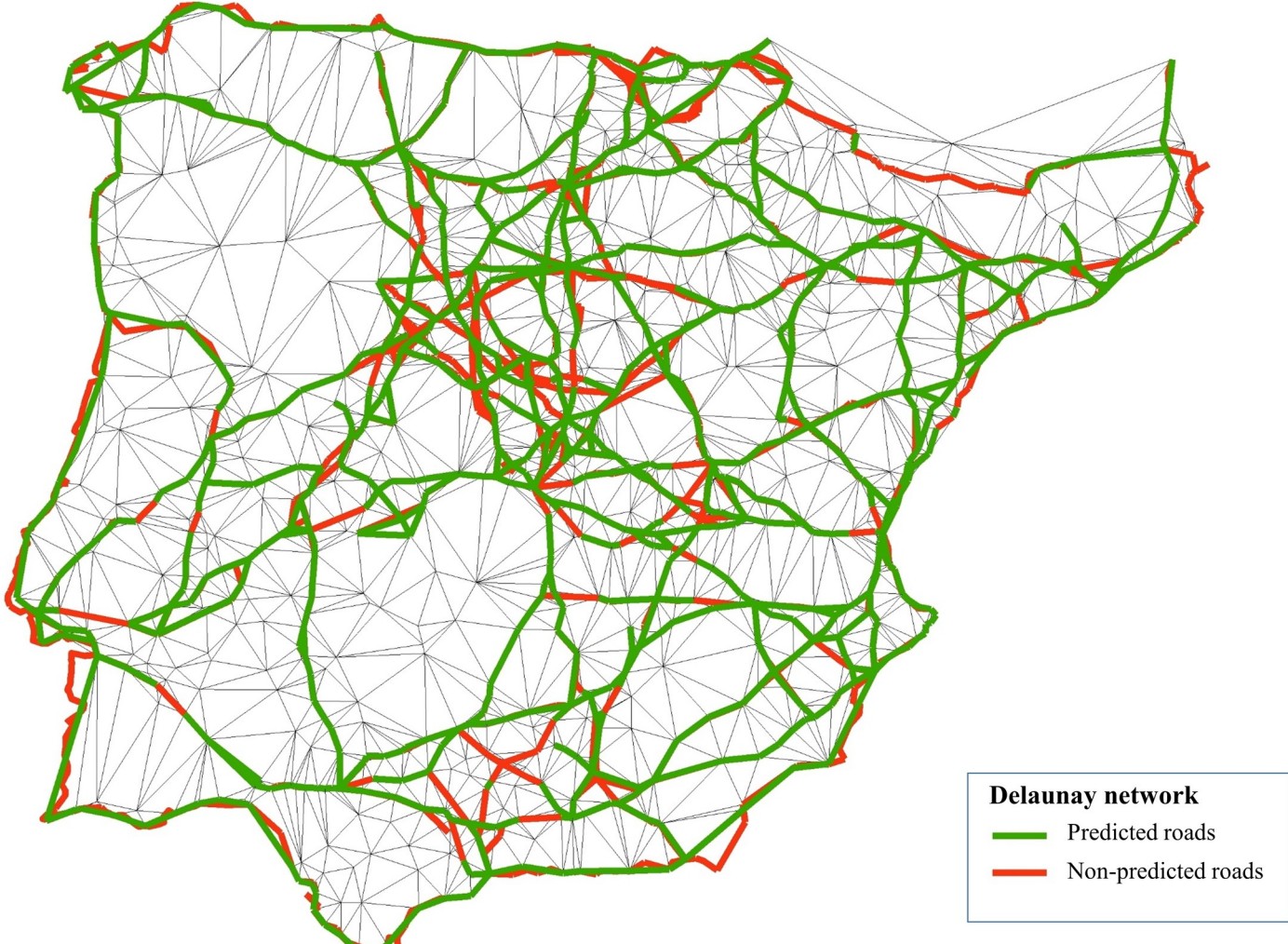

**Fig 8. Accuracy of the Delaunay network as predictor of Spanish early modern road network.** Notes: The sections of the historical roads effectively predicted by the Delaunay network are shown in green, and the errors are in red. Sources: Produced by the authors from D'Ocampo [107], Villuga [63], Estienne [108], Gail [109], Stella and L'Herba [110], Meneses [64], Rowlands [111], and Hogenberg [112]. Base map from the Centro Nacional de Información Geográfica (Spain).

also included, improving the efficiency and connectivity of the network. Some population settlements (inns, hermitages), whose exact locations were unknown, have not been included.

The paths that appear in the historical sources were adapted to the selected nodes to avoid duplication. This simplification ensures that a route that appears in several sources with slight differences does not give rise to several different paths.

Under these premises, the number of possible links between populations is 600,060, and the number of simplified links obtained from historical sources was 1,100. The generated Delaunay network consisted of 3,137 links, that is, it selected 0.52% of those possible. Despite being so restrictive, this selection was very effective because it coincided with 987 links from historical sources, 89.73% of cases. Note that the sections not predicted precisely, 10.27% of the cases, do not imply that the generated network is structurally different to the historical one since the wrong links in many cases are very close and practically parallel to those observed so that with a less strict criterion they could have been considered as coincidences.

The sections collected from historical sources that the Delaunay model could not generate were also incorporated to achieve a more complete network. The map titled "Spagna con le distantie de li loci" [122] was not included in the sources used because it does not accurately reflect roads but rather travel times between nearby cities (Annex 3 in S1 Annex).

**Lack of road categorization.** Although the lack of investment leading to poor road quality was not homogeneous, the real difference between roads was not related so much to the quality of the surface. It was linked somewhat to the least number of incidents expected during the trip. Lower numbers of incidents might have been due to the existence of bridges and inns, the repair of potholes, or greater security against attack. All these aspects of travel affected transport costs and therefore modified traffic flows and must be incorporated into the analysis.

Unfortunately, there is hardly any information available on the different road categories. The only itinerary where some information of this type appears is in Villuga [63], where a reference is made to an alternative road between Valladolid and Toledo which was more suitable for carriages due to its track width and lower slopes.

A notable exception is the Hispania map included in *Itinerarium Orbis Christiani*, by Hogenberg [142, 143], which is considered the first European atlas of roads [144–146] and which includes a classification of the roads by categories. This map presents a radial network that differs considerably from the decentralized one that appears in the Villuga [63] and Meneses [64] itineraries, where the relative use of each road is not taken into account. Six main roads radiate out from the center of the peninsula, and these coincide, to a large extent, with those which Ward [147] would propose almost two centuries later and which would serve as the basis for the design of the road network approved by Carlos III in 1761 [148].

This is a significant result because it indicates that the structure of the Spanish road network did not emerge ex novo with the arrival of the Bourbons after the War of Succession. It had already emerged by the end of the 16th century, with hardly any state support, as a direct result of the movement of people and goods. The 18th century road arbitrators, in reality, only formalized a road structure that had already existed de facto.

To confirm the reliability of the network structure shown by Hogenberg [112], it was compared to the itinerary information available for the early 17th century, under the hypothesis that the most critical roads would tend to appear in a more significant number of sources [18, 149] (Fig 9).

The results obtained confirm the reliability of the map of Hogenberg [112], showing a radial network centered on Toledo.

The Villuga and Meneses itineraries [63, 64] also provide some support for this radial version of the network if we consider the most repeated sections to be the most important or

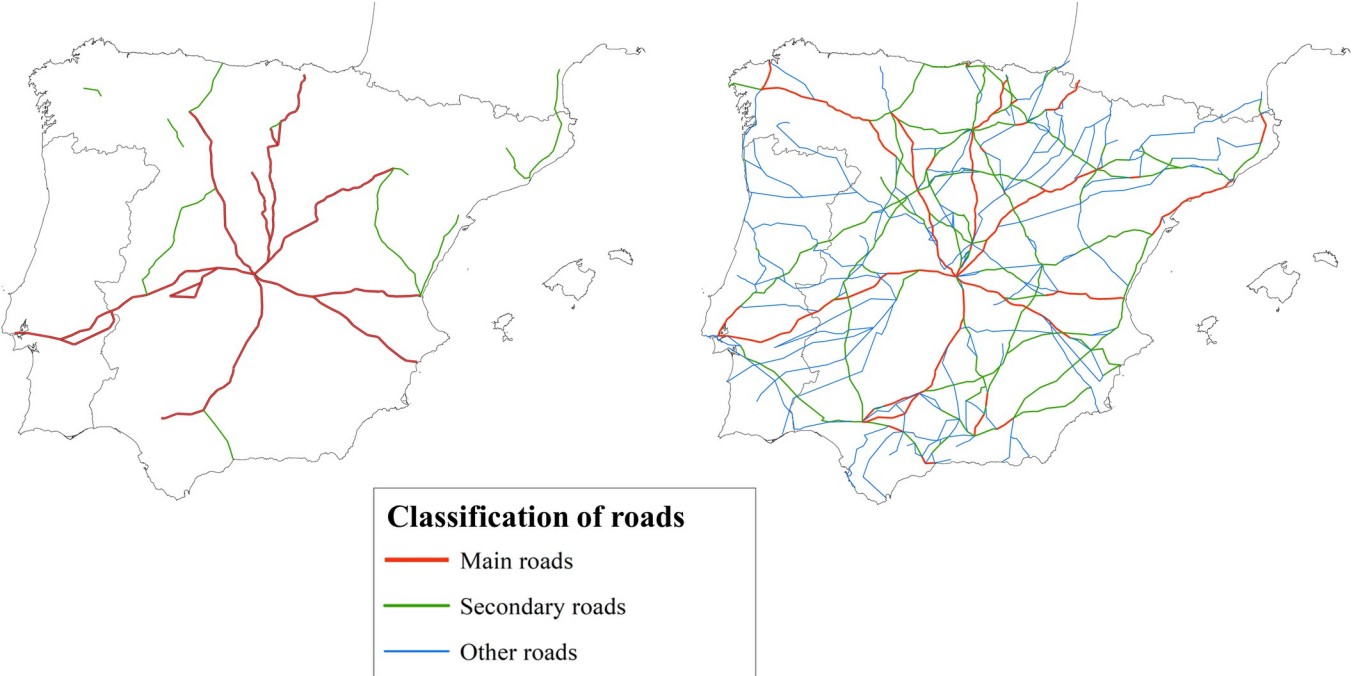

**Fig 9. Radial structure of the road network in the late sixteenth/early seventeenth centuries.** Notes: The categorization of the 1600 road network (right) was done by aggregating information from the three available sources for the period suggesting the methodology proposed by Menéndez Pidal [18] and Braudel [149]. In red are the main sections of road that appear in the three sources, in green are the secondary roads that appear in two, and blue are the less important roads that appear in a single source. *Source*: Produced by the authors from Hogenberg [112], Andree [150], Eichoviothias [151], and de Mayerne [152]. Base map from the Centro Nacional de Información Geográfica (Spain).

frequented (Fig 7). This puts Toledo at the center of the network, and, therefore, it is reasonable to assume that a large proportion of traffic passed through it [149].

The review of the dispersed available information confirms that the Villuga and Meneses itineraries [63, 64], although helpful, do not offer a reliable picture of the characteristics of the 16th century road network. This applies particularly to the categorization of roads, and they, therefore, cannot be used alone to analyze the logistic advantages of the different candidate cities for the capital.

To categorize the network sections, the same methodology was followed as before. According to the number of times the sections appear in the historical sources, they were classified into four groups—principal, primary, secondary, and others, obtaining a network structure very similar to that observed previously for the early 17th century, despite using different sources.

### Reconstructing the 16th century road network

**How the network was: The "real map".** The result of this process is the representation of the Spanish road network at the end of the 16th century shown in Fig 10.

This map coincides greatly with the current road map, which should not prove a surprise. The network's core is Toledo, as in the Menendez Pidal map (14), but the importance of other cities such as Burgos or Barcelona is not so prominent. The layout of the roads and the inclusion of slopes on transportation costs were performed using the methodology proposed by the authors for the 18th century road network [101].

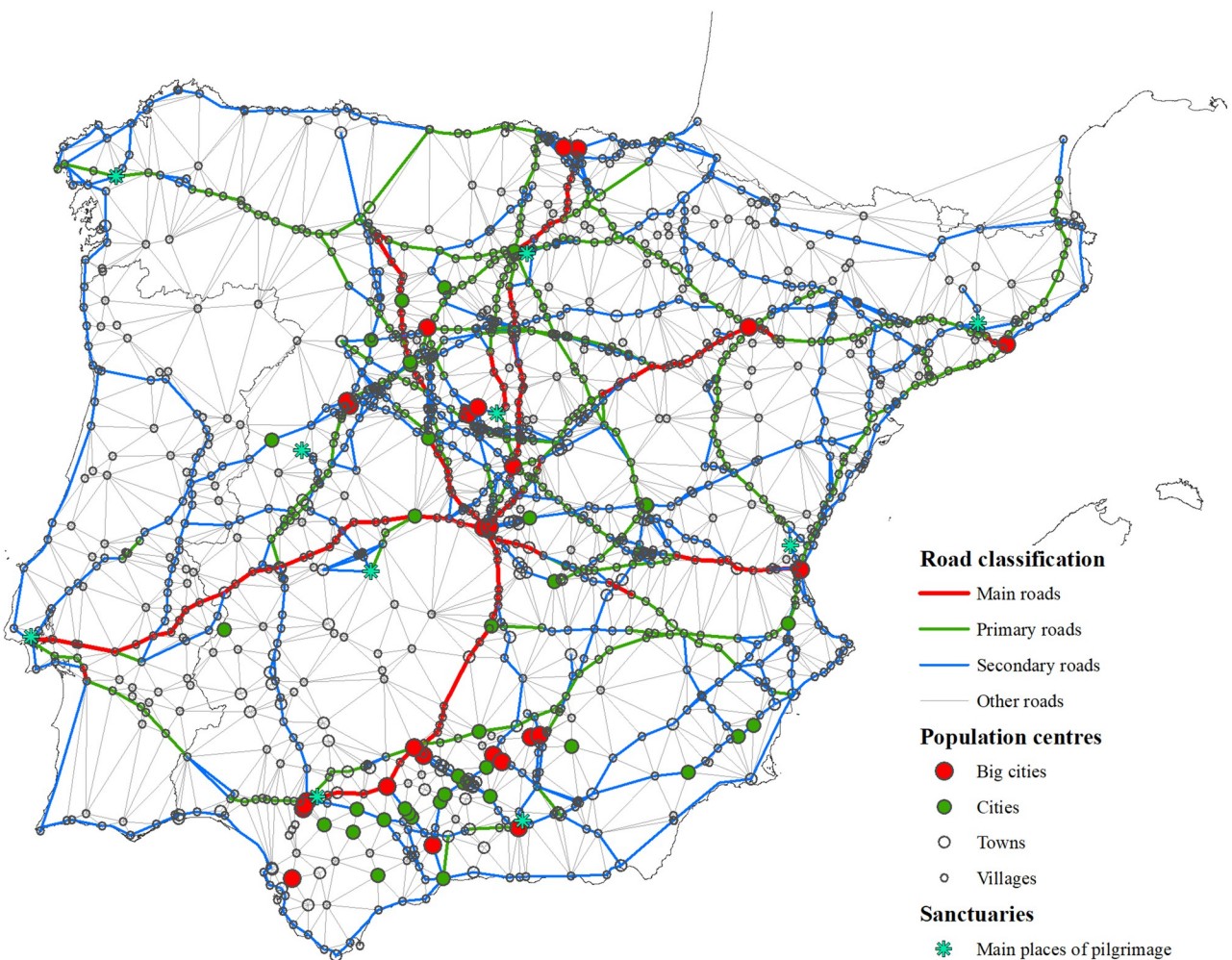

**Fig 10. The Iberian Road map of the late 16th century.** Notes: The categorization of the roads was done following the same methodology as in Fig 9. In red are the main sections of road that appear in at least six of the eight sources, in green the primary roads that appear in four or five, in blue the secondary roads that appear only in three or less, and finally in gray those artificially generated in the model and do not appear in any known source. *Sources*: Produced by the authors from D'Ocampo [107], Villuga [63], Estienne [108], Gail [109], Stella and L'Herba [110], Meneses [64], Rowlands [111], and Hogenberg [112]. Base map from the Centro Nacional de Información Geográfica (Spain).

**As they thought the network was: The "naïve map".**   The real road map of Spain in the 16<sup>th</sup> century is not enough to understand the parameters under which Philip II made his decision since the cartographic information he had in mind would have been erroneous to a significant extent. For this reason, in addition to the "real map", we have constructed another, transposing the information obtained about the roads to a 16<sup>th</sup> century map. This naïve map shows the network which people of the time would have had in their minds, according to their knowledge, and on which they would have based their decisions.

At this point, the choice of the reference map is an important issue due to the considerable advances in cartography during this period. Although cartographic development is generally considered an epiphenomenon concerning political change, in the 16<sup>th</sup> century, it played an active role in the formation of the modern state [153].

Given Philip II's interest in map making, it seems clear that he must have used, as the primary reference for his decision, some of the maps which emerged after the appearance of the map of Spain by Paletino Corsulensis [154]. In the years prior to the choice of the capital,

Philip II initiated two ambitious projects to provide Spain with a more precise and detailed map [155, 156]. However, both projects were abandoned before the appearance of the more accurate map of Paletino Corsulensis.

This map, printed in Venice, of which only one copy is available, was an important improvement on previous ones and was widely disseminated throughout Europe in the form of reduced copies, with slight variations being made by other authors such as Cock [157], Geminus [158], Luccini [159], Forlani [117]. This last map has been taken as a reference for this work since it has the issue date closest to when the capital city was chosen. However, any of the maps based on that of Paletino Corsulensis [154] would have given similar results [155, 160].

The "naïve map" of the late 16th century transport network was generated using a methodology analogous to that used to build the real network: all available itinerary information was incorporated, and the missing roads were generated through a Delaunay network connecting the populations which appear on the Forlani map. The categorization of the roads was carried out according to the frequency with which they appear in historical sources. The effect of the orography was introduced by raising the cost of transport on those sections which crossed the mountains on the map. Note that due to important errors in the geographic positioning of these mountains, new distortions were incorporated into the network.

As shown in Fig 11, the results obtained for the "naïve map" are much like those obtained for the real network.

## Advantages and disadvantages of candidate cities in terms of their impact on the whole territory: Network factors

Once the real and naïve transport networks were defined, complex network analysis was applied to determine their characteristics and to analyze the location advantages of those cities, which could have been chosen as the capital according to the previous historiographical analysis. The analysis shown below was limited to the road network and therefore did not include maritime traffic [101] due to the lack of precision of the available data. In any case, the results are robust to marine traffic levels such as those estimated in the second section.

### Most traveled cities: The betweenness centrality

In logistical terms, the betweenness centrality [161] of a town is the number of optimal paths that pass through it when all the network nodes are in contact with each other. It is an indicator that does not weigh the expected traffic since it does not consider the size of the towns, only their positioning in the network. Fig 12 (left) shows the betweenness centrality of the nodes of the transport network of the Iberian Peninsula in the second half of the 16th century.

Comparing these results with Villuga's [63] paths, where the sections are weighted by the frequency with which they appear (Fig 7), reveals the validity of the transport network generated. It also supports the hypothesis raised above; that the roads which appear in the itineraries are only a selection based on traffic and not an exhaustive list.

The remarkable similarity between the main roads, according to Villuga [63], and the betweenness centrality analysis is an indicator that in the 16th century, the primary determinant of traffic between towns and cities was their placement in the network rather than the size of their populations (Annex 4 in S1 Annex).

The betweenness centrality of the naïve transport network (Fig 12, right) coincides far less with Villuga's weighted itinerary. This indicates that its elaboration was not based on cartographic information but instead was carried out with direct input from travelers and merchants. It is surprising just how meticulously the information had to be selected to adequately reflect the intensity of the traffic between towns.

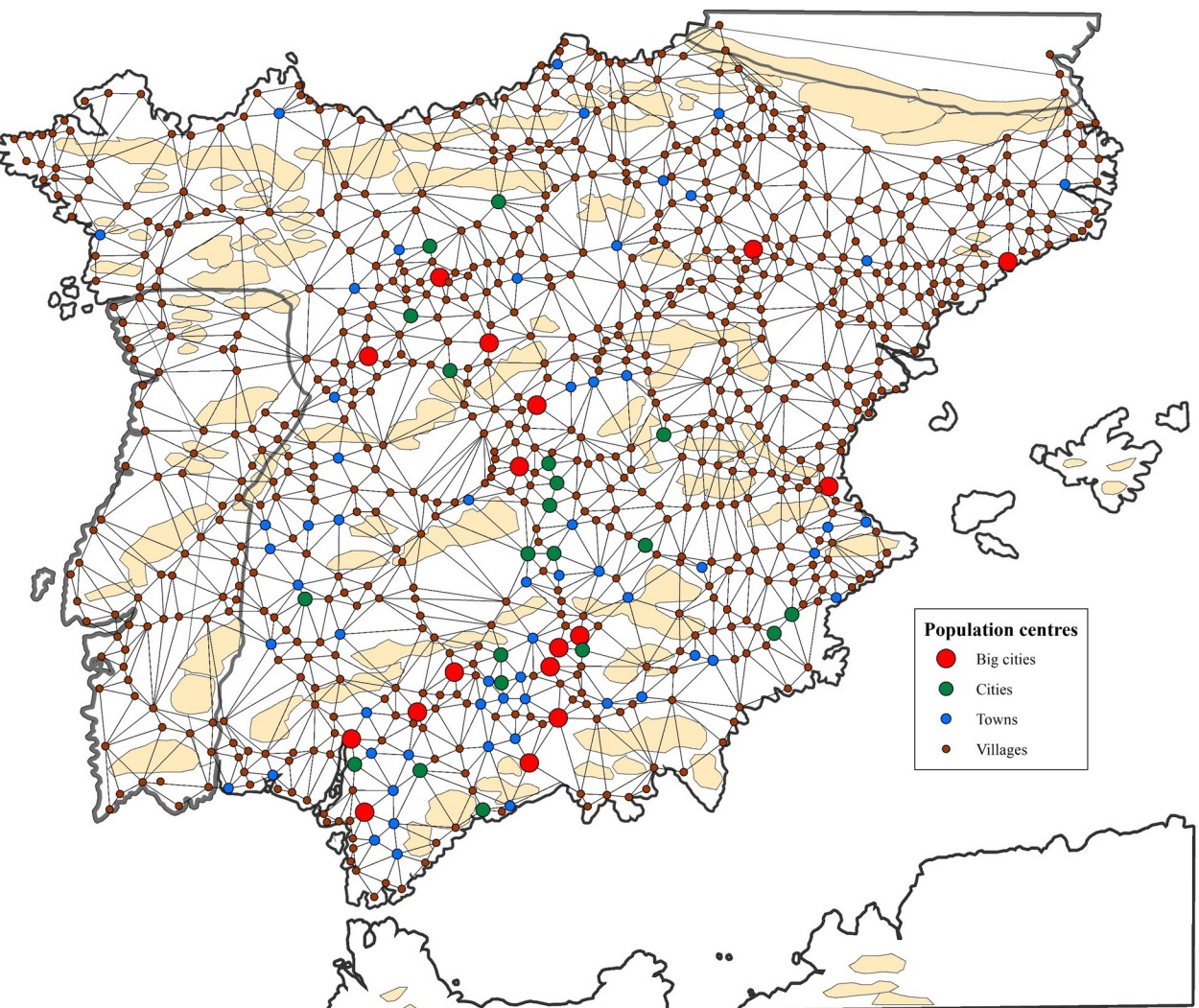

**Fig 11. The Iberian Road naïve map of the late 16th century.** Notes: The design of the network was made by adding to the available historical information the results obtained from the model in a similar way as in Fig 10, but now using the map of Forlani [117] as a spatial reference. Source: Produced by the authors from D'Ocampo [107], Villuga [63], Estienne [108], Gail [109], Stella and L'Herba [110], Meneses [64], Rowlands [111], and Hogenberg [112], and Forlani [117].

One of the most cited characteristics of Madrid to explain its choice as capital is its proximity to the graphical center of the peninsula. However, from a logistical point of view, proximity is not a property that can be defined solely from a geographical position. It depends on its relative position concerning other populations and the road network structure.

## The best-connected cities: The accessibility

The accessibility or closeness centrality of a town is defined as the inverse of the sum of transport costs to all other population settlements of the network through the optimal paths. Fig 13 shows the results obtained for the real and naïve transport networks. In the analysis carried out on the real network, it can be observed that Madrid is the city with the greatest closeness, followed at a short distance by Toledo. However, for the naïve network used for Philip II to

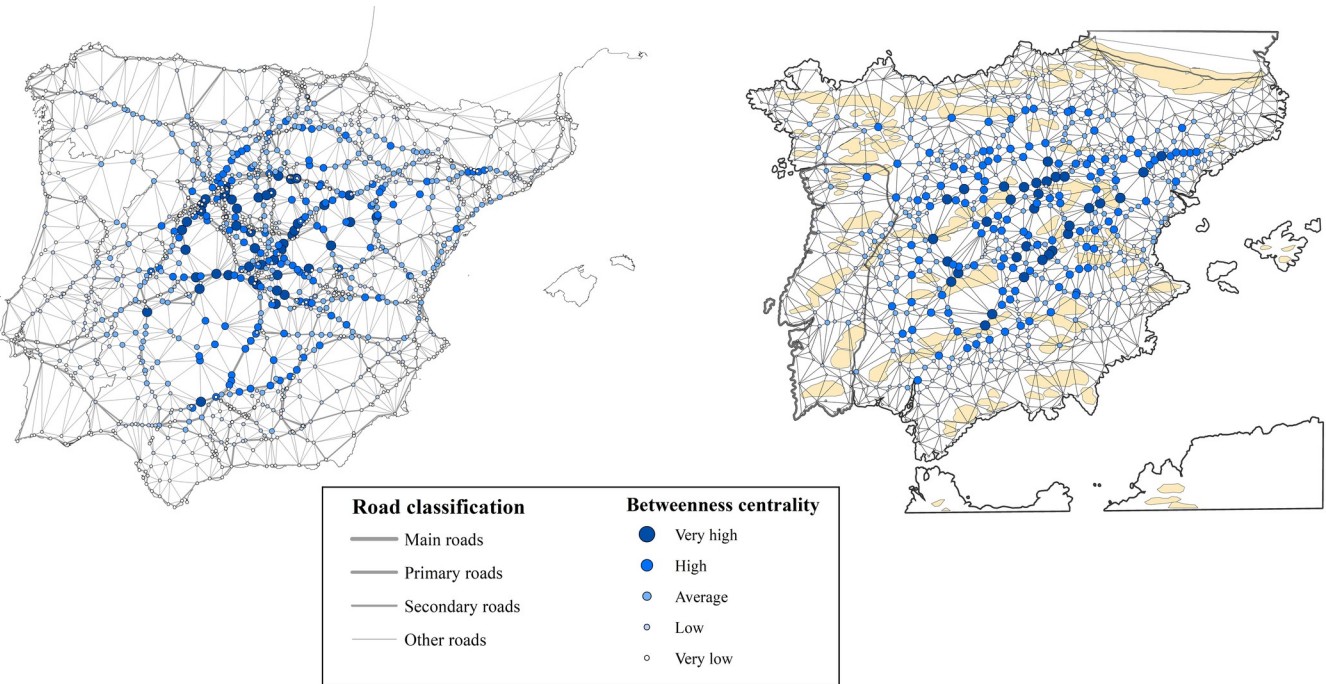

**Fig 12. Betweenness centrality maps.** *Notes*: The calculation of betweenness centrality does not consider road categories because they refer more to aspects of traffic than to the technical characteristics of roads. The length of the sections and their slope were included. *Source*: Produced by the authors from the networks shown in Figs 10 and 11. Base map from the Centro Nacional de Información Geográfica (Spain).

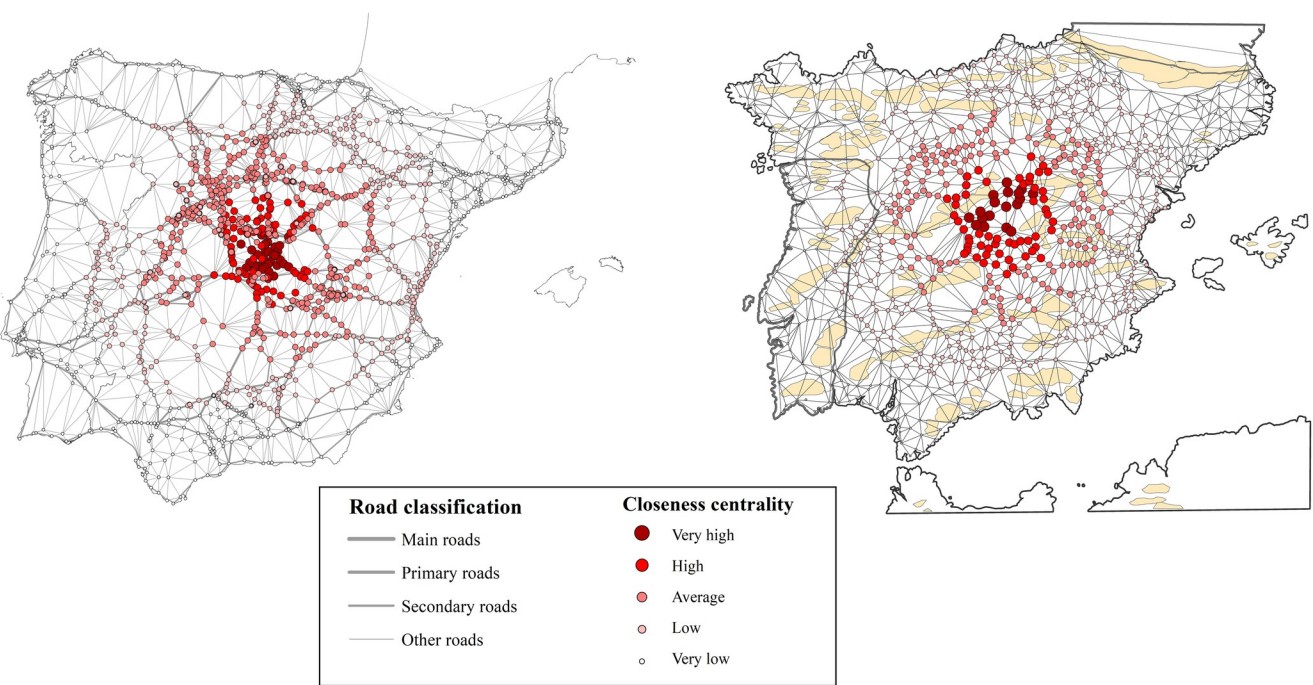

**Fig 13. Closeness centrality maps.** Notes: The calculation of closeness centrality does not take into account road categories because they refer more to aspects of traffic than to technical characteristics of roads. The length of the sections and their slope were included.

decide the capital, the area of greatest apparent closeness moves to the northeast, reducing Toledo's chances of becoming the capital but improving those of Barcelona and Zaragoza.

Predictably, this result coincides with the authors' results for the early 18[th] century [101].

## The most influential cities: The gravity model

Accessibility analysis faces severe limitations as it determines the places with the lowest transport costs but does not weigh the relative importance of the destinations. Gravity modeling allows us the take these into account through production or population.

To determine the suitability of different locations as the capital, the increase in traffic caused by the establishment of the capital in each of the main towns was calculated using this gravity model:

$$T_{ij} = \frac{\left(P_i P_j\right)^\alpha}{d_{ij}^\beta} \tag{1}$$

Where $T_{ij}$ is bilateral traffic between city $i$ and city $j$, $P_i$ is the population of city $i$, $P_j$ is the population of city $j$, and $d_{ij}$ is the minimum travel cost between these cities. As usual, the Dijkstra algorithm [162] was used to calculate the optimal paths between cities.

The primary determinant of transport cost is distance. Longer sections have a higher transport cost than shorter ones. To incorporate the difficulties of terrain, the transport cost has been multiplied by a ground roughness coefficient obtained from the number of crossed contour lines. Therefore, the expression for the cost of transport is the following:

$$c_{ij}^* = c_{ij}\left(1 + \gamma \frac{s_{ij}}{d_{ij}}\right) \tag{2}$$

where $c_{ij}^*$ is the final travel cost of the track including the orography effect, $c_{ij}$ is the Euclidean distance between the towns at the ends of the track, $\gamma$ is a coefficient that weights the effect of roughness, $s_{ij}$ is the number of contour lines crossed and $d_{ij}$ is the length of the track. This approach has the advantage of being independent of the traffic direction so that the network does not have to be bidirectional, and the transport costs from A to B are equal to those from B to A.

As the map of Forlani Veronese (1560) only incorporates qualitative information on the roughness of the terrain, the effect of orography was introduced into the "naïve map" by raising the cost of transport for those sections that cross the mountains which appear on it. Note that due to important errors in the geographic positioning of these mountains, new geographical misconceptions would have been created for Philip II and his advisors.

The possible variations in transport costs on different categories of roads, obtained in the previous section, were not considered, as they could have led to some bias in the results. However, it should be noted that its incorporation would have reinforced the results obtained.

The idea underlying this calculation is that a city is a good choice for the capital of a country if it increases the integration of markets, which implies maximizing the traffic with all the rest of the cities in the country. Since only those nodes were used for which detailed information on the number of inhabitants was available, 100 were used for the real network and 190 for the naïve network. Data on population comes from Correas [163], which refers to the year 1600 and is expressed in terms of numbers of taxpayers. The actual number of inhabitants was usually about four and a half times this figure, so that the total population of Madrid was around 34,000 inhabitants, very similar to that of Valladolid, while that of Toledo was nearly 50,000.

For our purposes, it was assumed that the population of the city which became the capital would increase in the medium term by 30,000 (the actual increase in the people of Madrid and Valladolid when each of them became the capital).

A complex procedure was followed to produce the estimates. Each city was considered in turn. Its population was increased by 30,000 as if it had become the capital, and the increase in bilateral traffic between it and every other city was calculated. So, for each town, 99 estimates were run with the "real map" and 189 with the naïve one.

Once the results for each town had been obtained, they were listed in order from that which generated the most significant increase of traffic to that which produced the slightest effect. Next, an index was developed from zero to one; zero corresponds to the worst choice for the capital from all other populations, and one corresponds to the best option.

As the historical information available does not allow us to determine with precision what the most suitable coefficients for the gravitational model should be, nine models were used with different values of α (1.0, 1.25, and 1.5) and β (1.6, 1.8, and 2.0) to simulate different types of traffic, varying the relevance of population size, on the one hand, and transport costs, on the other. Once obtained, the results were averaged to obtain a general result.

The final values for the main cities of the two networks are shown in Fig 14. The correlation between the two indicators is high (77%), which shows that the choice of capital based on the cartographic information of the 16th century was not in fact very biased.

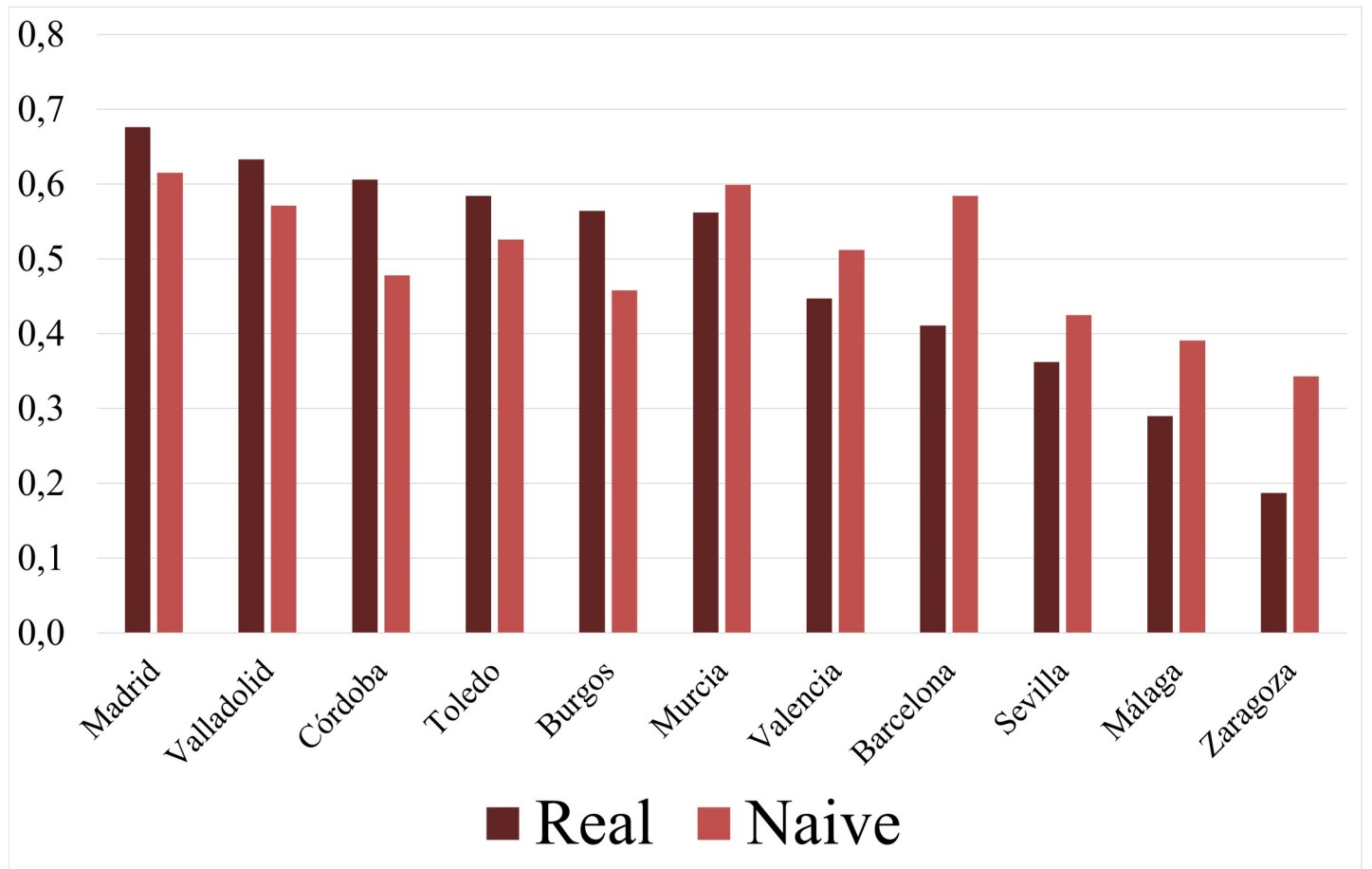

**Fig 14. Preferred locations for the capital.** Source: Produced by the authors from the networks shown in Figs 10 and 11 and populations from Correas [163].

[163] According to the naïve map, the city which best met the requirements to be the capital was Madrid, followed by Murcia, Barcelona and Valladolid.

With the "real map", the best option was also Madrid but now followed by Valladolid, Córdoba, and Toledo.

In any case, the results do not show, great differences. The criteria applied, rather than determining the specific city that should have been chosen, indicate the small group of cities that enjoyed the right conditions to become the capital of the Hispanic Monarchy. The final choice would depend on other economic and political determinants or even personal preferences of the king.

By the mid-16th century, Catalonia was slowly recovering from a succession of crises that began with the Black Death in the mid-14th century. These crises were first studied extensively by Vilar [164] and then discussed by several other authors as Feliu [165]. First, there was a demographic crisis with effects on agricultural production, then a financial one, and finally another on foreign trade. The civil war that started in 1462 would aggravate the economic situation. Although Barcelona withstood the effects of the demographic and agrarian crisis better than other parts of Catalonia, it was more affected by the foreign trade crisis and the civil war.

Murcia obtains a good result again, which is interesting because, as far as we know, this city has never been considered in the literature as a possible option for establishing the capital. However, it is not a particularly surprising result if you look at the data.

In the middle of the 16th century, Murcia had a population of some 17,000 inhabitants, much lower than other candidates. From a logistical point of view, it had easy access to Castile, Andalusia, and Aragon. It is also interesting to note that it belonged to the Crown of Castile, which would have largely eliminated possible misgivings regarding establishing the capital in another kingdom. Also, its proximity to the port of Cartagena, considered one of the safest in the Mediterranean [166], endowed it with many of the advantages of a coastal location, without the disadvantages.

Cartagena's position in the center of Spain's Mediterranean coastline put it in a much better place to become the hub of national cabotage trade than other more distant ports, such as Barcelona or Bilbao. It was favored by the growing orientation of Castile towards the Mediterranean during the second half of the 16th century. In addition, the shift toward the south of important maritime traffic routes to connect with those of the Atlantic also benefited the Murcia-Cartagena axis [167].

Seville's low ranking is notable, both in apparent and real terms, due to its difficult communication with Castile, which is not compensated for by its advantage in river transport.

Therefore, the network analysis shows that the choice of Madrid was justified from an economic point of view, given the importance of the Kingdom of Castile for Philip II; "the most important controlled unit of the Hapsburg's holding" (Ringrose [30], p. 761). This statement is valid both with the poor geographical information available in the 16th century and with today's maps.

However, the existence of other good alternatives suggests that other reasons were also considered; defensive, political, or personal.

From its real effects, the choice of Madrid can be considered reasonable or even positive for Spain as it was one of the options which allowed a more significant increase in passenger and freight traffic to be achieved. Moreover, its central position favored the development of a national structure under royal control, which would make it possible to overcome the fragmentation resulting from the growing power of cities [168] and establish a unified fiscal state, the main pending issue of the Hispanic monarchy during the 16th and 17th centuries [169, 170].

In addition, Madrid was the only city that could replace Toledo as the center of the road network without generating profound distortions in the existing transport flows [143]. Madrid's capital status only meant minor regional variations in the road network structure.

It is also important to note that the rapid growth experienced by Madrid during the second half of the 16th century was because of being the capital city itself, not due to the improvement of its road communication. In fact, in the next century after its designation as the capital, its population grew dramatically, while the road network was hardly altered.

These results are perfectly compatible with the fact that the predominance of urban and administrative activities and the high concentration of nobility in Madrid influenced its hinterland in some negative ways, being significantly damaging for Toledo. However, what would have happened in the case of Toledo, if it had continued as the capital of Spain, or if the choice had been Valladolid, is uncertain. It may well have been the case that the negative effect of rapid urban development would have arisen anyway.

## Conclusions

Philip II's choice of Madrid as his permanent residence and the imperial court in 1561 has had significant repercussions in the subsequent development of Spain's road network and infrastructure and, consequently, on the spatial configuration of economic activity.

Despite this, few quantitative studies have been conducted on the economic appropriateness of the choice made. However, very influential, indirect research has led to the idea that it was merely founded on political and religious reasoning, if not on personal whim. Moreover, it came at the cost of not choosing one of the coastal cities, particularly in the case of Barcelona, which developed more quickly later.

In this paper, the suitability of Madrid as the capital of Spain has been analyzed from a fundamentally economic perspective. The specific aim was to demonstrate that at the moment of being chosen as the capital, Madrid was at the core of economic life in the peninsula, as were Toledo and Valladolid.

To that end, we review the cost advantages of transport by ship over road transport. This is a key point as much of the criticism of the choice of Madrid as the capital has been based on the fact that it did not have a port. In the following, to analyze accurately the logistic issues we reconstructed the transport network map of Spain at the time using primary sources no used before supplemented by statistical methods. Additionally, we made a "naïve map", the one which King Philip II and his advisers would have had in mind, taking into account the limited knowledge of the age.

Based on these two maps, we analyzed the economic suitability of main Spanish cities as potential capitals through computational network methods.

The results based on the "naïve map" showed that the city that best met the requirements to be capital because of the effect it would have on the overall dynamics of road traffic on the peninsula was Madrid, closely followed by Murcia, Barcelona and Valladolid. The best option was also Madrid when the "real map" was considered, followed by Valladolid, Córdoba, and Toledo. Murcia remains in a relatively good position, but this is not the case for Barcelona. It is noticeable that Seville does not appear among the most suitable. Barcelona and Valencia have better rankings than Seville but not as good as Madrid, which consistently performs very well, clearly being the best option according to the results of both maps.

After a detailed review of the determining factors in the choice of Madrid as the capital, it can be affirmed that the decision was reasonable and even positive for Spain. Its advantageous position in the network of roads that emerged in the 16th century after the integration of the different peninsular kingdoms of the Hispanic Monarchy compensated to a great extent for

the disadvantages derived from not having communications by ship. In addition, its central position favored the expansion of trade and the development of a more solid national structure. For all these reasons, the choice of Madrid cannot consider a whim of Philip II since it had sound economic fundamentals.

Finally, we consider than the intensive use of the transport network as a key element in the analysis of historical events together with the application of complex network analysis opens a new and fruitful field of work in Cliometry. Moreover, we think that the explicit consideration of the inaccuracies of humanity's worldview before the 18th century could serve to reinterpret many historical events that are still not sufficiently explained.

## Supporting information

**S1 File.**
(RAR)

**S1 Annex.**
(DOCX)

## Author Contributions

**Conceptualization:** Rafael Myro.

**Data curation:** Federico Pablo-Martí, Ángel Alañón-Pardo.

**Formal analysis:** Federico Pablo-Martí.

**Funding acquisition:** Federico Pablo-Martí.

**Investigation:** Federico Pablo-Martí.

**Methodology:** Federico Pablo-Martí, Ángel Alañón-Pardo, Rafael Myro.

**Software:** Federico Pablo-Martí.

**Supervision:** Rafael Myro.

**Validation:** Federico Pablo-Martí.

**Writing – original draft:** Federico Pablo-Martí.

**Writing – review & editing:** Ángel Alañón-Pardo, Rafael Myro.

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
