## [Decision Letter · Decision Letter 0]

26 Oct 2021

PONE-D-21-29102Reality and perception in Madrid's location advantages as the capital of the Hispanic MonarchyPLOS ONE

Dear Dr. Pablo Martí,

Thank you for submitting your manuscript to PLOS ONE. After careful consideration, we feel that it has merit but does not fully meet PLOS ONE’s publication criteria as it currently stands. Therefore, we invite you to submit a revised version of the manuscript that addresses the points raised during the review process.

Both reviewers recommended major revision with many comments. Although both reviewers found interesting results from the study, they also pointed out drawbacks in quality of paper. I agree to their comments. I would like the authors to carefully check their comments and to improve the paper by addressing all of them.

We look forward to receiving your revised manuscript.

Kind regards,

Hironori Kato, Dr. Eng.

Academic Editor

PLOS ONE

“This work has been supported in part by Ministerio de Economía, Innovación y Competitividad (Spain) through Grants CSO2016-74888-C4-4-R (CITITALENT, AEI/FEDER, UE), Comunidad de Madrid H2019/HUM-5761 (INNJOBMAD-CM) and Universidad de Alcalá COVID-19 UAH 2019/00003/016/001/007.”

5. We note that Figure 1, 3, 5, 7, 9, 10, 11, A1, A2, A4, A5, A6 and A9 in your submission contain map images which may be copyrighted. All PLOS content is published under the Creative Commons Attribution License (CC BY 4.0), which means that the manuscript, images, and Supporting Information files will be freely available online, and any third party is permitted to access, download, copy, distribute, and use these materials in any way, even commercially, with proper attribution. For these reasons, we cannot publish previously copyrighted maps or satellite images created using proprietary data, such as Google software (Google Maps, Street View, and Earth). For more information, see our copyright guidelines: http://journals.plos.org/plosone/s/licenses-and-copyright.

a. You may seek permission from the original copyright holder Figure 1, 3, 5, 7, 9, 10, 11, A1, A2, A4, A5, A6 and A9 to publish the content specifically under the CC BY 4.0 license. 

6. Please ensure that you refer to Figure 6 and 7 in your text as, if accepted, production will need this reference to link the reader to the figure.

Reviewers' comments:

Reviewer's Responses to Questions

**Comments to the Author**

1. Is the manuscript technically sound, and do the data support the conclusions?

Reviewer #1: Partly

Reviewer #2: Partly

2. Has the statistical analysis been performed appropriately and rigorously? 

Reviewer #1: N/A

Reviewer #2: Yes

3. Have the authors made all data underlying the findings in their manuscript fully available?

Reviewer #1: Yes

Reviewer #2: Yes

4. Is the manuscript presented in an intelligible fashion and written in standard English?

Reviewer #1: Yes

Reviewer #2: Yes

5. Review Comments to the Author

Reviewer #1: The paper provides an interesting look at the decision of establishing a permanent capital and court in Madrid by Phillip II. It is interesting to read how logistical costs could have influenced the decision at that time. The authors also do an outstanding effort in reconstructing and gathering the data available at that time.

Still, there are some areas that could be improved (overall flow of the manuscript principally) and some that could be added to increase the value of the paper (for example discussion of the importance of the results for other studies).

1. The structure of the manuscript could be made more explicit by improving the division in sections.

a. Just by looking at the main sections (introduction -> The intrinsic factors -> The network factors -> The Spanish road network -> Results -> Conclusion) could better guide the reader through the manuscript (which is particularly important considering the length)

b. The introduction in its current form it is quite extensive. A shorter one would be preferable. As an example, “this paper aims to analyze Madrid’s suitability as Spain’s capital from an economic point of view” is in the seventh page.

c. In the page 9, the research is presented in several steps. However its clarity could be improved.

2. Title does not reflect the content

3. The abstract argues that “questioning the belief that this choice was eminently personal or political but lacked economic rationality”

a. Does the previous literature disregarded the economic merits of establishing the capital in Madrid?

i. There is a reference to Garcia Delgado (1990), which tells about (i) “unfavorable conditions for economic activity” (ii) “scarcity of primary resources”, (iii) large distance to coastal trading centers and lack of fluvial traffic”. It seems that the paper focus mainly on the (iii), which is fine, but if that is the case, it could be explicitly mentioned.

b. What does economic rationality mean in the context of this paper?

i. It seems that the rationale is more focused on the logistical properties and how that could have influenced the final choice among the different possibilities. Please clarify it.

4. It is mentioned in different parts the comparison with other alternatives. The implicit understanding is that those would have been considered at that time (although it is not sure if that is based on the actual historical knowledge or it is the assumption from the authors, which is fine but should be explained).

a. Furthermore, the alternatives analyzed seem to differ throughout the text. For example, in the introduction “Besides Madrid, other cities have been pointed out as possible locations, such as Valladolid, Toledo, Seville, or even Barcelona or Lisbon”. In Fig 1, other appear such as Malaga, Valencia, Murcia.

5. The paper employs several sources and methods to analyze each of the different variables. Whilst this is reasonable, it also adds some difficulty to its reading.

a. Page 9 “we analyze several Spanish cities’ suitability as potential capitals using a complex network approach”.

i. Complex network approach could be further explained here

6. The analysis of the three logistical costs (access to ports, fluvial traffic, and road transportation) could be followed by a comparison of these for all the cities considered (in reality or hypothetical based on the reasoning by the authors). It would be also interesting to include the potential differences in the weight of each in the final decision.

7. One concern that pop up in my mind through the text is why the logistical cost are estimated for the Spanish territory today and not others that were part of Phillip II reign. Please clarify.

8. It is not clear whether the paper looks to describe the decision-making process by Phillip II or to provide a general rationale that could have influenced any “rational” decision making.

a. If it is about Philipp II’s mainly, then the particularities and possible external influences on him could be included.

b. Or it could be to analyze how the logistical factors influenced his, based on the knowledge of the final decision.

c. Furthermore, it seems that all the factors considered had same weight (at least in logistics). Please clarify.

9. There is also some information that could be easily included to facilitate the reading by non-experts on this specific issue

a. Context of the establishment of the permanent capital

i. How it goes before? What was the historical context in which occurred?

ii. The benefits and drawbacks of a permanent capital

iii. What was the

iv. These are somehow described in different parts, but it could be strengthened

b. A more structured description of the different “theories” of Phillip II, including also some of the aspects that could have influenced him (role of advisors, importance of the lessons from his father, past experiences). Again, several of this are included in the manuscript, but these could be organized more clearly. It would be also upon of the reorganization of the manuscript (which is the main point to improve).

10. In page 44, it is mentioned that “according to the naïve map, the city which best met the requirements to be the capital was Madrid, followed by Valladolid, Murcia, and Barcelona”.

a. From, figure 12, it appears that whilst Madrid was first, second was Murcia, and almost equally Valladolid and Barcelona. Please check it.

b. Also, the difference seems to be very minimal. The authors could clarify how this small difference could have influenced the decision at that time (when similar methods and data employed by the authors were not available).

11. The conclusion could be

12. Internal references need to be revised

a. For example, in page 32, the reference is to Figure 2, please confirm if that is the adequate.

b. Similarly occurs with the reference to Figure 4 in page 38

13. The manuscript is related to previous work from the authors. This is acknowledged, but from a personal point of view, it could be explicitly mentioned as “in a previous study, the authors…” (or similar sentence). That is a matter of personal preference, so I have no strong opinion on it.

Reviewer #2: This paper aims at revisiting the choice of Madrid by Felipe II de España as the capital based on limited knowledge at the time, especially in terms of the economic validity. It gives us not only deeper understanding of the Spanish land development history but also interesting insights and argument for the future development. Some further verification would however be necessary to have good grounds for estimating the economic soundness of Madrid as the capital. Moreover, I'm afraid that some parts of the article might be difficult to follow and the English writings could be improved. Overall, I'd recommend this paper for the journal for its unique findings of academic and practical importance with additional validation of the conclusion and careful editing of the English writings.

(1) Further validation of the conclusion

If I understand correctly, the choice of Felipe II is assumed to be based on limited knowledge, namely, the proposed naïve map in this research. It would be true that Madrid had a good reason to be chosen as the capital for its economic advantages, even with limited information (the "naïve map"), and that choice could be assessed as economically rational with the present knowledge (based on the "real map"). But I don't see if the gaps of the index scores between Madrid and other options, especially Valladolid, Murcia and Barcelona, are so significant that we can be very confident that Felipe II took the economic validity into consideration in the choice of capital. I'd suggest that further examination of data and argument be made to elaborate the evaluation of the 'candidate' cities based on your network analysis so that you can conclude Madrid was "clearly (...) the best option" (p.46).

Valladolid and Madrid had relative advantages in (then perceived) economy, like Murcia and Barcelona, based on your network analysis. And Madrid would have been more competitive in terms of the intrinsic characteristics. Madrid also had the predominance of urban and administrative activities and the high concentration of nobility, which can justify the choice perhaps politically, practically and financially. But again, it would not be easy to conclude Madrid was the best choice solely from the results of the network analysis.

(2) Polishing the English writings

There would be many sentences in the paper that are not easy to follow. (For instance, "However, it looks strange for Madrid's suitability from an economic perspective not to have been taken into account by Philip II and his advisers. However, despite the importance of the issue, it has attracted little direct analysis" (p.6). The first sentence could be differently written for easier understanding/ The repetition of 'however' should be avoided. In the same page (p.6), another sentence which requires some improvement would be "According to such suggestions, Madrid's main advantage, from Philip II's point of view, was probably that neither the Church nor the nobility enjoyed an important presence there, which would facilitate the development of a new administration, under the solely under royal supervision.")

Also, many repeated contents are found; for instance, the conclusion section has too many repeated sentences of the previous section. Such simple repetition should be removed, and your thorough editing, or restructuring, of the paper would also be recommended to make the paper more efficiently written, that is, reader-friendly. A proofreading by native English speaker, preferably a professional, should be performed.

(3) Possible further discussion on the findings

Among the four major options for the capital (Madrid, Valladolid, Murcia and Barcelona), especially from the restricted assessment at that time (the "naïve" map), only Barcelona seems to have a rather big gap between the assessment based on the "real" map and that based on the "naïve" one. In what way can we interpret its relatively low evaluation based on the "real" map, compared to the subsequent economic and urban development? If your interpretation on this issue about Barcelona affects the conclusion on Madrid in some way, it could be added to your paper.

(4) Minor issues

* p.4: "Thus, his sister, Juana of Austria, regent between 1554 and 1556, and between 1556 and 1559, she told him in August 1958 that she preferred Madrid to Valladolid as the capital, or, failing that, Guadalajara or Burgos". -> It should be August 1558, not 1958.

* p.5: "Therefore, the establishment of a permanent capital meant the renunciation of the Carolinian practice of the itinerant monarchy, a course which, with all its drawbacks, had the great advantage of occasionally giving its peoples visible proof that their king had not forgotten them". -> It should be people, not peoples.

* p.6: "However, this does not mean that it has not been commented on and discussed a great deal, with religious and political reasons often cited as to why Madrid was more suitable than, say, Valladolid Toledo, Seville, or even Barcelona or Lisbon". -> "," is missing between "Valladolid" and "Toledo".

* p.7: ""The very existence of Madrid as a capital city was the consequence of a political decision. No other city in early modern Europe was as dependent upon administered economic life, and no major city was so poorly located to stimulate market-oriented exchanges" (Ringrose 1983, p. 8)". -> "Europe" should be in italic.

* p.8: "Madrazo Madrazo (1984b) attempted to identify the infrastructure work carried out under the Bourbons". -> In some parts, authors are referred with its first and second surnames, while they have only the first surname in others. Better to be coherent?

* p.9: "According to data from Bairoch et al. (1988), at the beginning of the 16th century, the most populated cities in Spain were located in the south (Seville and Malaga) and in the east (Valencia), all three having more than 40.00 inhabitants". -> It should be 40,000, not 40.00. Also, punctuation of three digits (thousand) should be coherent throughout the paper, preferably comma in English.

* p.11: "It did not have enough houses and its location on a hillock and surrounded by the River Tajo made it difficult to build more. r(Alvar Ezquerra 1985)" -> There is an unnecessary "r". And, the reference information should be before the period.

* p.11: "Although it may seem strange to us now, the exposition to epidemics was a question of crucial importance, mainly talking about the capital location where the high number of travelers passing through was an element of added risk". -> "(E)xposition" should be exposure, I guess. And, there are too many short paragraphs in this page.

* p.18: "3. The logistical advantages of the port cities depend on the latency of the network and the goods transported". -> "3" should be in black color.

** I don't raise all those points to be revised. There seem to be a lot... Please have a thorough proofreading.

* P.20: I don't well understand what "Estimated year data" means in Fig. 3.

* P.35: "The review of the somewhat disjointed information available confirms that the Villuga and Meneses itineraries, although useful, do not offer a reliable picture of the characteristics of the sixteenth-century road network". -> What is "somewhat disjointed information"? A bit more careful expression would be appreciated.

6. PLOS authors have the option to publish the peer review history of their article (what does this mean?). If published, this will include your full peer review and any attached files.

Reviewer #1: No

Reviewer #2: **Yes: **Shunsaku Komatsuzaki

---

## [Author Response · Author response to Decision Letter 0]

18 Feb 2022

The rebuttal letter is in the Attach files.

We would like to thank the editor and both reviewers for giving us the possibility to improve our work. We believe that the comments made are very constructive and appropriate. Their incidence on the expository weaknesses of the initial version of the article has not helped us to make a new one that we hope they consider substantially better.

---

## [Decision Letter · Decision Letter 1]

1 Apr 2022

PONE-D-21-29102R1The choice of Madrid as the capital of Spain by Philip II in the light of the knowledge of his time: a transport network perspectivePLOS ONE

Dear Dr. Pablo Martí,

Thank you for submitting your manuscript to PLOS ONE. After careful consideration, we feel that it has merit but does not fully meet PLOS ONE’s publication criteria as it currently stands. Therefore, we invite you to submit a revised version of the manuscript that addresses the points raised during the review process.

The recommendations from Reviewers 1 and 2 are accept and major revision, respectively. Reviewer 2 raises the issues regarding the quality of language and chapter structure. I also agree to the comments. I would like the authors to improve the paper by addressing them.

We look forward to receiving your revised manuscript.

Kind regards,

Hironori Kato, Dr. Eng.

Academic Editor

PLOS ONE

Journal Requirements:

Reviewers' comments:

Reviewer's Responses to Questions

**Comments to the Author**

1. If the authors have adequately addressed your comments raised in a previous round of review and you feel that this manuscript is now acceptable for publication, you may indicate that here to bypass the “Comments to the Author” section, enter your conflict of interest statement in the “Confidential to Editor” section, and submit your "Accept" recommendation.

Reviewer #1: All comments have been addressed

Reviewer #2: (No Response)

2. Is the manuscript technically sound, and do the data support the conclusions?

Reviewer #1: Yes

Reviewer #2: Yes

3. Has the statistical analysis been performed appropriately and rigorously? 

Reviewer #1: N/A

Reviewer #2: N/A

4. Have the authors made all data underlying the findings in their manuscript fully available?

Reviewer #1: Yes

Reviewer #2: Yes

5. Is the manuscript presented in an intelligible fashion and written in standard English?

Reviewer #1: Yes

Reviewer #2: No

6. Review Comments to the Author

Reviewer #1: (No Response)

Reviewer #2: Your thorough revision responding to our comments is highly appreciated. I see huge improvements, especially in the framing of your study (objectives and conclusions) which now matches better your data and analysis. However, there remain two major issues to be addressed in your paper.

(1) The quality of English

I'd say that the quality of English is not enough yet for recommending to an international journal with readers from all over the world. There are a lot of sentences with a nest of too many clauses/phrases. Readers would find it very difficult to follow. Particularly, the first and second chapters ("Introduction" and "Advantages and disadvantages of candidate cities considered on their own") should be re-edited. It is highly recommended that you get advice from native English speaker, not only about grammar but also about reader friendly writing.

(An example in p.4: "Elliot (10) also pointed out that Philip II was mistaken in believing that the decision to reside in the geometrical center of the peninsula would produce the impression of absolute impartiality in the treatment accorded to his subjects, since, although this was not the intention of Philip II, the very choice of the capital in the heart of Castile granted his government a Castilian color".)

(Another example in p.8: "From the point of view of urban functionality, the narrow streets of Toledo, still organized along medieval lines, did not allow the court to move with adequate splendor and ease, making it difficult to ride on horseback through many of the streets and preventing movement by carriage along most of them (44)".)

In addition, there remain many typographical errors and some grammar issues.

Please look at my comments on the attached manuscript.

Without seeing a significant improvement in writing, especially error correction, it would be difficult for me to recommend your paper to the journal. Again, your very careful proofreading would be strongly advised.

(2) The position/meaning of the analysis of intrinsic characteristics in the whole structure

As your abstract and conclusion imply, the network analysis with using the two maps (one of your novel contributions) would be primary and the analysis of intrinsic characteristics would be supplementary. The position of the second chapter ("Advantages and disadvantages of candidate cities considered on their own") would thus look unclear in the whole structure. It can perhaps be discussed after the network analysis.

If you have any clear reason to put this chapter here (right after the introduction and before the network analysis), the logical flow between this chapter and the next should be improved. The structure of the conclusion could also be reconsidered.

7. PLOS authors have the option to publish the peer review history of their article (what does this mean?). If published, this will include your full peer review and any attached files.

Reviewer #1: **Yes: **Daniel del Barrio Alvarez

Reviewer #2: **Yes: **Shunsaku Komatsuzaki

---

## [Author Response · Author response to Decision Letter 1]

12 May 2022

Thank you again for your kind and thoughtful comments. We have tried to respond to them adequately. In particular, we have enlisted the help of a company specialized in editing scientific papers. We hope that the text will now be more understandable.

Regarding the comments on the structure, we have made some small changes but we have kept the general outline. The objective has been to maintain the habitual scheme in the works on the capital of Madrid. The unusual length and structure of the article is due to the need to justify in great detail the assertions made about the advantages of road transport as they represent a very different view from the current one. We believe that with the improvements introduced during the editing process the work will be of interest to a wide audience.

---

## [Editor Report · Decision Letter 2]

23 May 2022

The choice of Madrid as the capital of Spain by Philip II in the light of the knowledge of his time: a transport network perspective

PONE-D-21-29102R2

Dear Dr. Pablo Martí,

We’re pleased to inform you that your manuscript has been judged scientifically suitable for publication and will be formally accepted for publication once it meets all outstanding technical requirements.

Kind regards,

Hironori Kato, Dr. Eng.

Academic Editor

PLOS ONE

---

## [Editor Report · Acceptance letter]

30 May 2022

PONE-D-21-29102R2 

The choice of Madrid as the capital of Spain by Philip II in the light of the knowledge of his time: a transport network perspective 

Dear Dr. Pablo-Martí:

I'm pleased to inform you that your manuscript has been deemed suitable for publication in PLOS ONE. Congratulations! Your manuscript is now with our production department. 

Kind regards, 

on behalf of

Dr. Hironori Kato 

Academic Editor

PLOS ONE